# Star-Shaped Denoising Diffusion Probabilistic Models

**Andrey Okhotin**[*]
HSE University, MSU University
Moscow, Russia
andrey.okhotin@gmail.com

**Dmitry Molchanov**[*]
BAYESG
Budva, Montenegro
dmolch111@gmail.com

**Vladimir Arkhipkin**
Sber AI
Moscow, Russia
arkhipkin.v98@gmail.com

**Grigory Bartosh**
AMLab, Informatics Institute
University of Amsterdam
Amsterdam, Netherlands
g.bartosh@uva.nl

**Viktor Ohanesian**
Independent Researcher
v.v.oganesyan@gmail.com

**Aibek Alanov**
AIRI, HSE University
Moscow, Russia
alanov.aibek@gmail.com

**Dmitry Vetrov**
Constructor University
Bremen, Germany
dvetrov@constructor.university

## Abstract

Denoising Diffusion Probabilistic Models (DDPMs) provide the foundation for the recent breakthroughs in generative modeling. Their Markovian structure makes it difficult to define DDPMs with distributions other than Gaussian or discrete. In this paper, we introduce Star-Shaped DDPM (SS-DDPM). Its *star-shaped diffusion process* allows us to bypass the need to define the transition probabilities or compute posteriors. We establish duality between star-shaped and specific Markovian diffusions for the exponential family of distributions and derive efficient algorithms for training and sampling from SS-DDPMs. In the case of Gaussian distributions, SS-DDPM is equivalent to DDPM. However, SS-DDPMs provide a simple recipe for designing diffusion models with distributions such as Beta, von Mises–Fisher, Dirichlet, Wishart and others, which can be especially useful when data lies on a constrained manifold. We evaluate the model in different settings and find it competitive even on image data, where Beta SS-DDPM achieves results comparable to a Gaussian DDPM. Our implementation is available at https://github.com/andrey-okhotin/star-shaped.

## 1 Introduction

Deep generative models have shown outstanding sample quality in a wide variety of modalities. Generative Adversarial Networks (GANs) (Goodfellow et al., 2014; Karras et al., 2021), autoregressive models (Ramesh et al., 2021), Variational Autoencoders (Kingma & Welling, 2013; Rezende et al., 2014), Normalizing Flows (Grathwohl et al., 2018; Chen et al., 2019) and energy-based models (Xiao et al., 2020) show impressive abilities to synthesize objects. However, GANs are not robust to the choice of architecture and optimization method (Arjovsky et al., 2017; Gulrajani et al., 2017; Karras et al., 2019; Brock et al., 2018), and they often fail to cover modes in data distribution (Zhao et al., 2018; Thanh-Tung & Tran, 2020). Likelihood-based models avoid mode collapse but may overestimate the probability in low-density regions (Zhang et al., 2021).

---

[*]Equal contribution

37th Conference on Neural Information Processing Systems (NeurIPS 2023).

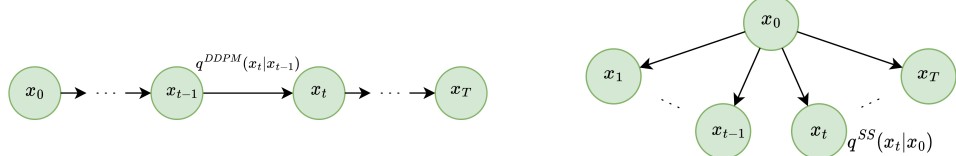

Figure 1: Markovian forward processes of DDPM (left) and the star-shaped forward process of SS-DDPM (right).

Recently, diffusion probabilistic models (Sohl-Dickstein et al., 2015; Ho et al., 2020) have received a lot of attention. These models generate samples using a trained Markov process that starts with white noise and iteratively removes noise from the sample. Recent works have shown that diffusion models can generate samples comparable in quality or even better than GANs (Song et al., 2020b; Dhariwal & Nichol, 2021), while they do not suffer from mode collapse by design, and also they have a log-likelihood comparable to autoregressive models (Kingma et al., 2021). Moreover, diffusion models show these results in various modalities such as images (Saharia et al., 2021), sound (Popov et al., 2021; Liu et al., 2022) and shapes (Luo & Hu, 2021; Zhou et al., 2021).

The main principle of diffusion models is to destroy information during the forward process and then restore it during the reverse process. In conventional diffusion models like denoising diffusion probabilistic models (DDPM) destruction of information occurs through the injection of Gaussian noise, which is reasonable for some types of data, such as images. However, for data distributed on manifolds, bounded volumes, or with other features, the injection of Gaussian noise can be unnatural, breaking the data structure. Unfortunately, it is not clear how to replace the noise distribution within traditional diffusion models. The problem is that we have to maintain a connection between the distributions defining the Markov noising process that gradually destroys information and its marginal distributions. While some papers explore other distributions, such as delta functions (Bansal et al., 2022) or Gamma distribution (Nachmani et al., 2021), they provide ad hoc solutions for special cases that are not easily generalized.

In this paper, we present Star-Shaped Denoising Diffusion Probabilistic Models (SS-DDPM), a new approach that generalizes Gaussian DDPM to an exponential family of noise distributions. In SS-DDPM, one only needs to define marginal distributions at each diffusion step (see Figure 1). We provide a derivation of SS-DDPM, design efficient sampling and training algorithms, and show its equivalence to DDPM (Ho et al., 2020) in the case of Gaussian noise. Then, we outline a number of practical considerations that aid in training and applying SS-DDPMs. In Section 5, we demonstrate the ability of SS-DDPM to work with distributions like von Mises–Fisher, Dirichlet and Wishart. Finally, we evaluate SS-DDPM on image and text generation. Categorical SS-DDPM matches the performance of Multinomial Text Diffusion (Hoogeboom et al., 2021) on the `text8` dataset, while our Beta diffusion model achieves results, comparable to a Gaussian DDPM on CIFAR-10.

## 2 Theory

### 2.1 DDPMs

We start with a brief introduction of DDPMs. The Gaussian DDPM (Ho et al., 2020) is defined as a forward (diffusion) process $q^{\mathrm{DDPM}}(x_{0:T})$ and a corresponding reverse (denoising) process $p_\theta^{\mathrm{DDPM}}(x_{0:T})$. The forward process is defined as a Markov chain with Gaussian conditionals:

$$q^{\mathrm{DDPM}}(x_{0:T}) = q(x_0)\prod_{t=1}^{T} q^{\mathrm{DDPM}}(x_t|x_{t-1}), \tag{1}$$

$$q^{\mathrm{DDPM}}(x_t|x_{t-1}) = \mathcal{N}\left(x_t; \sqrt{1-\beta_t}x_{t-1}, \beta_t\mathbf{I}\right), \tag{2}$$

where $q(x_0)$ is the data distribution. Parameters $\beta_t$ are typically chosen in advance and fixed, defining the noise schedule of the diffusion process. The noise schedule is chosen in such a way that the final $x_T$ no longer depends on $x_0$ and follows a standard Gaussian distribution $q^{\mathrm{DDPM}}(x_T) = \mathcal{N}\left(x_T; 0, \mathbf{I}\right)$.

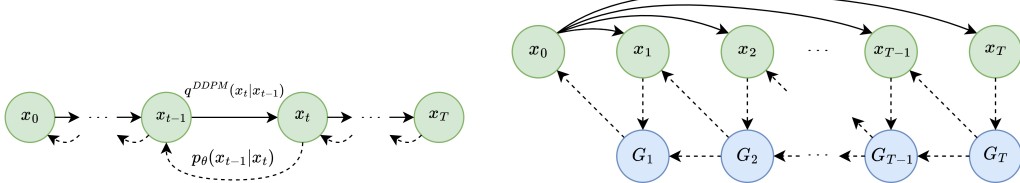

(a) Denoising Diffusion Probabilistic Models    (b) Star-Shaped Denoising Diffusion Probabilistic Models

Figure 2: Model structure of DDPM and SS-DDPM.

The reverse process $p_\theta^{\text{DDPM}}(x_{0:T})$ follows a similar structure and constitutes the generative part of the model:

$$p_\theta^{\text{DDPM}}(x_{0:T}) = q^{\text{DDPM}}(x_T) \prod_{t=1}^{T} p_\theta^{\text{DDPM}}(x_{t-1}|x_t), \tag{3}$$

$$p_\theta^{\text{DDPM}}(x_{t-1}|x_t) = \mathcal{N}\left(x_{t-1}; \mu_\theta(x_t, t), \Sigma_\theta(x_t, t)\right). \tag{4}$$

The forward process $q^{\text{DDPM}}(x_{0:T})$ of DDPM is typically fixed, and all the parameters of the model are contained in the generative part of the model $p_\theta^{\text{DDPM}}(x_{0:T})$. These parameters are tuned to maximize the variational lower bound (VLB) on the likelihood of the training data:

$$\mathcal{L}^{\text{DDPM}}(\theta) = \mathbb{E}_{q^{\text{DDPM}}} \left[ \log p_\theta^{\text{DDPM}}(x_0|x_1) - \sum_{t=2}^{T} D_{KL}\left(q^{\text{DDPM}}(x_{t-1}|x_t, x_0) \,\|\, p_\theta^{\text{DDPM}}(x_{t-1}|x_t)\right) \right]$$
$$\tag{5}$$

$$\mathcal{L}^{\text{DDPM}}(\theta) \to \max_\theta \tag{6}$$

One of the main challenges in defining DDPMs is the computation of the posterior $q^{\text{DDPM}}(x_{t-1}|x_t, x_0)$. Specifically, the transition probabilities $q^{\text{DDPM}}(x_t|x_{t-1})$ have to be defined in such a way that this posterior is tractable. Specific DDPM-like models are available for Gaussian (Ho et al., 2020), Categorical (Hoogeboom et al., 2021) and Gamma (Kawar et al., 2022) distributions. Defining such models remains challenging in more general cases.

## 2.2 Star-Shaped DDPMs

As previously discussed, extending the DDPMs to other distributions poses significant challenges. In light of these difficulties, we propose to construct a model that only relies on marginal distributions $q(x_t|x_0)$ in its definition and the derivation of the loss function.

We define star-shaped diffusion as a *non-Markovian* forward process $q^{\text{SS}}(x_{0:T})$ that has the following structure:

$$q^{\text{SS}}(x_{0:T}) = q(x_0) \prod_{t=1}^{T} q^{\text{SS}}(x_t|x_0), \tag{7}$$

where $q(x_0)$ is the data distribution. We note that in contrast to DDPM all noisy variables $x_t$ are conditionally independent given $x_0$ instead of constituting a Markov chain. This structure of the forward process allows us to utilize other noise distributions, which we discuss in more detail later.

## 2.3 Defining the reverse model

In DDPMs the true reverse model $q^{\text{DDPM}}(x_{0:T})$ has a Markovian structure (Ho et al., 2020), allowing for an efficient sequential generation algorithm:

$$q^{\text{DDPM}}(x_{0:T}) = q^{\text{DDPM}}(x_T) \prod_{t=1}^{T} q^{\text{DDPM}}(x_{t-1}|x_t). \tag{8}$$

For the star-shaped diffusion, however, the Markovian assumption breaks:

$$q^{\text{SS}}(x_{0:T}) = q^{\text{SS}}(x_T) \prod_{t=1}^{T} q^{\text{SS}}(x_{t-1}|x_{t:T}). \tag{9}$$

Consequently, we now need to approximate the true reverse process by a parametric model which is conditioned on the whole tail $x_{t:T}$.

$$p_\theta^{\text{SS}}(x_{0:T}) = p_\theta^{\text{SS}}(x_T) \prod_{t=1}^{T} p_\theta^{\text{SS}}(x_{t-1}|x_{t:T}). \tag{10}$$

It is crucial to use the whole tail $x_{t:T}$ rather than just one variable $x_t$ when predicting $x_{t-1}$ in a star-shaped model. As we show in Appendix B, if we try to approximate the true reverse process with a Markov model, we introduce a substantial irreducible gap into the variational lower bound. Such a sampling procedure fails to generate realistic samples, as can be seen in Figure 3.

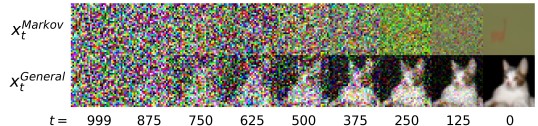

Figure 3: Markov reverse process fails to recover realistic images from star-shaped diffusion, while a general reverse process produces realistic images. The top row is equivalent to DDIM at $\sigma_t^2 = 1 - \alpha_{t-1}$. A similar effect was also observed by Bansal et al. (2022).

Intuitively, in DDPMs the information about $x_0$ that is contained in $x_{t+1}$ is nested into the information about $x_0$ that is contained in $x_t$. That is why knowing $x_t$ allows us to discard $x_{t+1}$. In star-shaped diffusion, however, all variables contain independent pieces of information about $x_0$ and should all be taken into account when making predictions.

We can write down the variational lower bound as follows:

$$\mathcal{L}^{\text{SS}}(\theta) = \mathbb{E}_{q^{\text{SS}}} \left[ \log p_\theta(x_0|x_{1:T}) - \sum_{t=2}^{T} D_{KL} \left( q^{\text{SS}}(x_{t-1}|x_0) \,\|\, p_\theta^{\text{SS}}(x_{t-1}|x_{t:T}) \right) \right] \tag{11}$$

With this VLB, we only need the marginal distributions $q(x_{t-1}|x_0)$ to define and train the model, which allows us to use a wider variety of noising distributions. Since conditioning the predictive model $p_\theta(x_{t-1}|x_{t:T})$ on the whole tail $x_{t:T}$ is typically impractical, we propose a more efficient way to implement the reverse process next.

## 2.4 Efficient tail conditioning

Instead of using the full tail $x_{t:T}$, we would like to define some statistic $G_t = \mathcal{G}_t(x_{t:T})$ that would extract all information about $x_0$ from the tail $x_{t:T}$. Formally speaking, we call $G_t$ a *sufficient tail statistic* if the following equality holds:

$$q^{\text{SS}}(x_{t-1}|x_{t:T}) = q^{\text{SS}}(x_{t-1}|G_t). \tag{12}$$

One way to define $G_t$ is to concatenate all the variables $x_{t:T}$ into a single vector. This, however, is impractical, as its dimension would grow with the size of the tail $T - t + 1$.

The Pitman–Koopman–Darmois (Pitman, 1936) theorem (PKD) states that exponential families admit a sufficient statistic with constant dimensionality. It also states that no other distribution admits one: if such a statistic were to exist, the distribution has to be a member of the exponential family. Inspired by the PKD, we turn to the exponential family of distributions. In the case of star-shaped diffusion, we cannot apply the PKD directly, as it was formulated for i.i.d. samples and our samples are not identically distributed. However, we can still define a sufficient tail statistic $G_t$ for a specific subset of the exponential family, which we call an *exponential family with linear parameterization*:

**Theorem 1.** *Assume the forward process of a star-shaped model takes the following form:*

$$q^{\text{SS}}(x_t|x_0) = h_t(x_t) \exp \left\{ \eta_t(x_0)^\mathsf{T} \mathcal{T}(x_t) - \Omega_t(x_0) \right\}, \tag{13}$$

$$\eta_t(x_0) = A_t f(x_0) + b_t. \tag{14}$$

*Let $G_t$ be a tail statistic, defined as follows:*

$$G_t = \mathcal{G}_t(x_{t:T}) = \sum_{s=t}^{T} A_s^\mathsf{T} \mathcal{T}(x_s). \tag{15}$$

*Then, $G_t$ is a sufficient tail statistic:*

$$q^{\text{ss}}(x_{t-1}|x_{t:T}) = q^{\text{ss}}(x_{t-1}|G_t). \tag{16}$$

Here definition (13) is the standard definition of the exponential family, where $h_t(x_t)$ is *the base measure*, $\eta_t(x_0)$ is the vector of *natural parameters* with corresponding *sufficient statistics* $\mathcal{T}(x_t)$, and $\Omega_t(x_0)$ is *the log-partition function*. The key assumption added is the *linear parameterization* of the natural parameters (14). We provide the proof in Appendix C. When $A_t$ is scalar, we denote it as $a_t$ instead.

For the most part, the premise of Theorem 1 restricts the parameterization of the distributions rather than the family of the distributions involved. As we discuss in Appendix F, we found it easy to come up with linear parameterization for a wide range of distributions in the exponential family. For example, we can obtain a linear parameterization for the Beta distribution $q(x_t|x_0) = \text{Beta}(x_t; \alpha_t, \beta_t)$ using $x_0$ as the mode of the distribution and introducing a new concentration parameter $\nu_t$:

$$\alpha_t = 1 + \nu_t x_0, \tag{17}$$
$$\beta_t = 1 + \nu_t(1 - x_0). \tag{18}$$

In this case, $\eta_t(x_0) = \nu_t x_0$, $\mathcal{T}(x_t) = \log \frac{x_t}{1-x_t}$, and we can use equation (15) to define the sufficient tail statistic $G_t$. We provide more examples in Appendix F. We also provide an implementation-ready reference sheet for a wide range of distributions in the exponential family in Table 6.

We suspect that, just like in PKD, this trick is only possible for a subset of the exponential family. In the general case, the dimensionality of the sufficient tail statistic $G_t$ would have to grow with the size of the tail $x_{t:T}$. It is still possible to apply SS-DDPM in this case, however, crafting the (now only approximately) sufficient statistic $G_t$ would require more careful consideration and we leave it for future work.

## 2.5 Final model definition

To maximize the VLB (11), each step of the reverse process should approximate the true reverse distribution:

$$p_\theta^{\text{ss}}(x_{t-1}|x_{t:T}) \approx q^{\text{ss}}(x_{t-1}|x_{t:T}) = \int q^{\text{ss}}(x_{t-1}|x_0)q^{\text{ss}}(x_0|x_{t:T})dx_0. \tag{19}$$

Similarly to DDPM (Ho et al., 2020), we choose to approximate $q^{\text{ss}}(x_0|x_{t:T})$ with a delta function centered at the prediction of some model $x_\theta(\mathcal{G}_t(x_{t:T}), t)$. This results in the following definition of the reverse process of SS-DDPM:

$$p_\theta^{\text{ss}}(x_{t-1}|x_{t:T}) = q^{\text{ss}}(x_{t-1}|x_0)\big|_{x_0 = x_\theta(\mathcal{G}_t(x_{t:T}), t)}. \tag{20}$$

The distribution $p_\theta^{\text{ss}}(x_0|x_{1:T})$ can be fixed to some small-variance distribution $p_\theta^{\text{ss}}(x_0|\hat{x}_0)$ centered at the final prediction $\hat{x}_0 = x_\theta(\mathcal{G}_1(x_{1:T}), 1)$, similar to the dequantization term, commonly used in DDPM. If this distribution has no trainable parameters, the corresponding term can be removed from the training objective. This dequantization distribution would then only be used for log-likelihood estimation and, optionally, for sampling.

Together with the forward process (7) and the VLB objective (11), this concludes the general definition of the SS-DDPM model. The model structure is illustrated in Figure 2. The corresponding training and sampling algorithms are provided in Algorithms 1 and 2.

The resulting model is similar to DDPM in spirit. We follow the same principles when designing the forward process: starting from a low-variance distribution, centered at $x_0$ at $t = 1$, we gradually increase the entropy of the distribution $q^{\text{ss}}(x_t|x_0)$ until there is no information shared between $x_0$ and $x_t$ at $t = T$.

We provide concrete definitions for Beta, Gamma, Dirichlet, von Mises, von Mises–Fisher, Wishart, Gaussian and Categorical distributions in Appendix F.

| **Algorithm 1** SS-DDPM training | **Algorithm 2** SS-DDPM sampling |
|---|---|
| **repeat** $\quad x_0 \sim q(x_0)$ $\quad t \sim \text{Uniform}(1, \ldots, T)$ $\quad x_{t:T} \sim q^{\text{ss}}(x_{t:T}|x_0)$ $\quad G_t = \sum_{s=t}^{T} A_s^{\mathsf{T}} \mathcal{T}(x_s)$ $\quad$ Move along $\nabla_\theta \text{KL}(q^{\text{ss}}(x_{t-1}|x_0) \| p_\theta^{\text{ss}}(x_{t-1}|G_t))$ **until** Convergence | $x_T \sim q^{\text{ss}}(x_T)$ $G_T = A_T^{\mathsf{T}} \mathcal{T}(x_T)$ **for** $t = T$ to $2$ **do** $\quad \tilde{x}_0 = x_\theta(G_t, t)$ $\quad x_{t-1} \sim q^{\text{ss}}(x_{t-1}|x_0)\big|_{x_0 = \tilde{x}_0}$ $\quad G_{t-1} = G_t + A_{t-1}^{\mathsf{T}} \mathcal{T}(x_{t-1})$ **end for** $x_0 \sim p_\theta^{\text{ss}}(x_0|G_1)$ |

## 2.6 Duality between star-shaped and Markovian diffusion

While the variables $x_{1:T}$ follow a star-shaped diffusion process, the corresponding tail statistics $G_{1:T}$ form a Markov chain:

$$G_t = \sum_{s=t}^{T} A_s^{\mathsf{T}} \mathcal{T}(x_s) = G_{t+1} + A_t^{\mathsf{T}} \mathcal{T}(x_t), \tag{21}$$

since $x_t$ is conditionally independent from $G_{t+2:T}$ given $G_{t+1}$ (see Appendix E for details). Moreover, we can rewrite the probabilistic model in terms of $G_t$ and see that variables $(x_0, G_{1:T})$ form a (not necessarily Gaussian) DDPM.

In the case of Gaussian distributions, this duality makes SS-DDPM and DDPM equivalent. This equivalence can be shown explicitly:

**Theorem 2.** *Let $\overline{\alpha}_t^{\text{DDPM}}$ define the noising schedule for a DDPM model (1–2) via $\beta_t = (\overline{\alpha}_{t-1}^{\text{DDPM}} - \overline{\alpha}_t^{\text{DDPM}})/\overline{\alpha}_{t-1}^{\text{DDPM}}$. Let $q^{\text{ss}}(x_{0:T})$ be a Gaussian SS-DDPM forward process with the following noising schedule and sufficient tail statistic:*

$$q^{\text{ss}}(x_t|x_0) = \mathcal{N}\left(x_t; \sqrt{\overline{\alpha}_t^{\text{ss}}} x_0, 1 - \overline{\alpha}_t^{\text{ss}}\right), \tag{22}$$

$$\mathcal{G}_t(x_{t:T}) = \frac{1 - \overline{\alpha}_t^{\text{DDPM}}}{\sqrt{\overline{\alpha}_t^{\text{DDPM}}}} \sum_{s=t}^{T} \frac{\sqrt{\overline{\alpha}_s^{\text{ss}}} x_s}{1 - \overline{\alpha}_s^{\text{ss}}}, \text{ where} \tag{23}$$

$$\frac{\overline{\alpha}_t^{\text{ss}}}{1 - \overline{\alpha}_t^{\text{ss}}} = \frac{\overline{\alpha}_t^{\text{DDPM}}}{1 - \overline{\alpha}_t^{\text{DDPM}}} - \frac{\overline{\alpha}_{t+1}^{\text{DDPM}}}{1 - \overline{\alpha}_{t+1}^{\text{DDPM}}}. \tag{24}$$

*Then the tail statistic $G_t$ follows a Gaussian DDPM noising process $q^{\text{DDPM}}(x_{0:T})|_{x_{1:T}=G_{1:T}}$ defined by the schedule $\overline{\alpha}_t^{\text{DDPM}}$. Moreover, the corresponding reverse processes and VLB objectives are also equivalent.*

We show this equivalence in Appendix D. We make use of this connection when choosing the noising schedule for other distributions.

This equivalence means that SS-DDPM is a direct generalization of Gaussian DDPM. While admitting the Gaussian case, SS-DDPM can also be used to implicitly define a non-Gaussian DDPM in the space of sufficient tail statistics for a wide range of distributions.

## 3 Practical considerations

While the model is properly defined, there are several practical considerations that are important for the efficiency of star-shaped diffusion.

**Choosing the right schedule** It is important to choose the right noising schedule for a SS-DDPM model. It significantly depends on the number of diffusion steps $T$ and behaves differently given different noising schedules, typical to DDPMs. This is illustrated in Figure 4, where we show the noising schedules for Gaussian SS-DDPMs that are equivalent to DDPMs with the same cosine schedule.

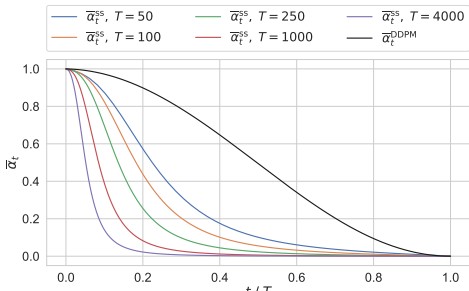

Figure 4: The noising schedule $\overline{\alpha}_t^{\mathrm{SS}}$ for Gaussian star-shaped diffusion, defined for different numbers of steps $T$ using eq. (24). All the corresponding equivalent DDPMs have the same cosine schedule $\overline{\alpha}_t^{\mathrm{DDPM}}$.

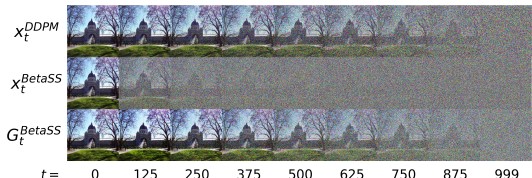

Figure 5: Top: samples $x_t$ from a Gaussian DDPM forward process with a cosine noise schedule. Bottom: samples $G_t$ from a Beta SS-DDPM forward process with a noise schedule obtained by matching the mutual information. Middle: corresponding samples $x_t$ from that Beta SS-DDPM forward process. The tail statistics have the same level of noise as $x_t^{\mathrm{DDPM}}$, while the samples $x_t^{\mathrm{BetaSS}}$ are diffused much faster.

Since the variables $G_t$ follow a DDPM-like process, we would like to somehow reuse those DDPM noising schedules that are already known to work well. For Gaussian distributions, we can transform a DDPM noising schedule into the corresponding SS-DDPM noising schedule analytically by equating $I(x_0; G_t) = I(x_0; x_t^{\mathrm{DDPM}})$. In general case, we look for schedules that have approximately the same level of mutual information $I(x_0; G_t)$ as the corresponding mutual information $I(x_0; x_t^{\mathrm{DDPM}})$ for a DDPM model for all timesteps $t$. We estimate the mutual information using Kraskov (Kraskov et al., 2004) and DSIVI (Molchanov et al., 2019) estimators and build a look-up table to match the noising schedules. This procedure is described in more detail in Appendix G. The resulting schedule for the Beta SS-DDPM is illustrated in Figure 5. Note how with the right schedule appropriately normalized tail statistics $G_t$ look and function similarly to the samples $x_t$ from the corresponding Gaussian DDPM. We further discuss this in Appendix H.

**Implementing the sampler** During sampling, we can grow the tail statistic $G_t$ without any overhead, as described in Algorithm 2. However, during training, we need to sample the tail statistic for each object to estimate the loss function. For this we need to sample the full tail $x_{t:T}$ from the forward process $q^{\mathrm{SS}}(x_{t:T}|x_0)$, and then compute the tail statistic $G_t$. In practice, this does not add a noticeable overhead and can be computed in parallel to the training process if needed.

**Reducing the number of steps** We can sample from DDPMs more efficiently by skipping some timestamps. This wouldn't work for SS-DDPM, because changing the number of steps would require changing the noising schedule and, consequently, retraining the model.

However, we can still use a similar trick to reduce the number of function evaluations. Instead of skipping some timestamps $x_{t_1+1:t_2-1}$, we can draw them from the forward process using the current prediction $x_\theta(G_{t_2}, t_2)$, and then use these samples to obtain the tail statistic $G_{t_1}$. For Gaussian SS-DDPM this is equivalent to skipping these timestamps in the corresponding DDPM. In general case, it amounts to approximating the reverse process with a different reverse process:

$$p_\theta^{\mathrm{SS}}(x_{t_1:t_2}|G_{t_2}) = \prod_{t=t_1}^{t_2} q^{\mathrm{SS}}(x_t|x_0)\big|_{x_0=x_\theta(G_t,t)} \approx \prod_{t=t_1}^{t_2} q^{\mathrm{SS}}(x_t|x_0)\big|_{x_0=x_\theta(G_{t_2},t_2)}. \qquad (25)$$

We observe a similar dependence on the number of function evaluations for SS-DDPMs and DDPMs.

**Time-dependent tail normalization** As defined in Theorem 1, the tail statistics can have vastly different scales for different timestamps. The values of coefficients $a_t$ can range from thousandths when $t$ approaches $T$ to thousands when $t$ approaches zero. To make the tail statistics suitable for use in neural networks, proper normalization is crucial. In most cases, we collect the time-dependent means and variances of the tail statistics across the training dataset and normalize the tail statistics to zero mean and unit variance. We further discuss this issue in Appendix H.

**Architectural choices** To make training the model easier, we make some minor adjustments to the neural network architecture and the loss function.

Our neural networks $x_\theta(G_t, t)$ take the tail statistic $G_t$ as an input and are expected to produce an estimate of $x_0$ as an output. In SS-DDPM the data $x_0$ might lie on some manifold, like the unit sphere or the space of positive definite matrices. Therefore, we need to map the neural network output to that manifold. We do that on a case-by-case basis, as described in Appendices I–L.

Different terms of the VLB can have drastically different scales. For this reason, it is common practice to train DDPMs with a modified loss function like $L_{simple}$ rather than the VLB to improve the stability of training (Ho et al., 2020). Similarly, we can optimize a reweighted variational lower bound when training SS-DDPMs.

## 4 Related works

Our work builds upon Denoising Diffusion Probabilistic Models (Ho et al., 2020). Interest in diffusion models has increased recently due to their impressive results in image (Ho et al., 2020; Song et al., 2020b; Dhariwal & Nichol, 2021) and audio (Popov et al., 2021; Liu et al., 2022) generation.

SS-DDPM is most closely related to DDPM. Like DDPM, we only rely on variational inference when defining and working with our model. SS-DDPM can be seen as a direct generalization of DDPM and essentially is a recipe for defining DDPMs with non-Gaussian distributions. The underlying non-Gaussian DDPM is constructed implicitly and can be seen as dual to the star-shaped formulation that we use throughout the paper.

Other ways to construct non-Gaussian DDPMs include Binomial diffusion (Sohl-Dickstein et al., 2015), Multinomial diffusion (Hoogeboom et al., 2021) and Gamma diffusion (Nachmani et al., 2021). Each of these works presents a separate derivation of the resulting objective, and extending them to other distributions is not straightforward. On the other hand, SS-DDPM provides a single recipe for a wide range of distributions.

DDPMs have several important extensions. Song et al. (2020a) provide a family of non-Markovian diffusions that all result in the same training objective as DDPM. One of them, denoted Denoising Diffusion Implicit Model (DDIM), results in an efficient deterministic sampling algorithm that requires a much lower number of function evaluations than conventional stochastic sampling. Their derivations also admit a star-shaped forward process (at $\sigma_t^2 = 1 - \alpha_{t-1}$), however, the model is not studied in the star-shaped regime. Their reverse process remains Markovian, which we show to not be sufficient to invert a star-shaped forward process. Denoising Diffusion Restoration Models (Kawar et al., 2022) provide a way to solve general linear inverse problems using a trained DDPM. They can be used for image restoration, inpainting, colorization and other conditional generation tasks. Both DDIMs and DDRMs rely on the explicit form of the underlying DDPM model and are derived for Gaussian diffusion. Extending these models to SS-DDPMs is a promising direction for future work.

Song et al. (2020b) established the connection between DDPMs and models based on score matching. This connection gives rise to continuous-time variants of the models, deterministic solutions and more precise density estimation using ODEs. We suspect that a similar connection might hold for SS-DDPMs as well, and it can be investigated further in future works.

Other works that present diffusion-like models with other types of noise or applied to manifold data, generally stem from score matching rather than variational inference. Flow Matching (Lipman et al., 2022) is an alternative probabilistic framework that works with any differentiable degradation process. De Bortoli et al. (2022) and Huang et al. (2022) extended score matching to Riemannian manifolds, and Chen & Lipman (2023) proposed Riemannian Flow Matching. Bansal et al. (2022) proposed Cold Diffusion, a non-probabilistic approach to reversing general degradation processes.

To the best of our knowledge, for the first time, an approach for constructing diffusion without a consecutive process was proposed by Rissanen et al. (2022) (IHDM) and further expanded on by Daras et al. (2022) and Hoogeboom & Salimans (2022). IHDM uses a similar star-shaped structure that results in a similar variational lower bound. Adding a deterministic process based on the heat equation allows the authors to keep the reverse process Markovian without having to introduce the tail statistics. As IHDM heavily relies on blurring rather than adding noise, the resulting diffusion dynamics become very different. Conceptually our work is much closer to DDPM than IHDM.

# 5 Experiments

We evaluate SS-DDPM with different families of noising distributions. The experiment setup, training details and hyperparameters are listed in Appendices I–L.

**Synthetic data**  We consider two examples of star-shaped diffusion processes with Dirichlet and Wishart noise to generate data from the probabilistic simplex and from the manifold of p.d. matrices respectively. We compare them to DDPM, where the predictive network $x_\theta(x_t, t)$ is parameterized to always satisfy the manifold constraints. As seen in Table 1, using the appropriate distribution results in a better approximation. The data and modeled distributions are illustrated in Table 3. This shows the ability of SS-DDPM to work with different distributions and generate data from exotic domains.

Table 1: KL divergence between the real data distribution and the model distribution $D_{KL}\left(q(x_0) \,\|\, p_\theta(x_0)\right)$ for Gaussian DDPM and SS-DDPM.

|         | Dirichlet | Wishart |
|---------|-----------|---------|
| DDPM    | 0.200     | 0.096   |
| SS-DDPM | **0.011** | **0.037** |

**Geodesic data**  We apply SS-DDPM to a geodesic dataset of fires on the Earth's surface (EOSDIS, 2020) using a three-dimensional von Mises–Fisher distribution. The resulting samples and the source data are illustrated in Table 3. We find that SS-DDPM is not too sensitive to the distribution family and can fit data in different domains.

**Discrete data**  Categorical SS-DDPM is similar to Multinomial Text Diffusion (MTD, (Hoogeboom et al., 2021)). However, unlike in the Gaussian case, these models are not strictly equivalent. We follow a similar setup to MTD and apply Categorical SS-DDPM to unconditional text generation on the `text8` dataset (Mahoney, 2011). As shown in Table 2, SS-DDPM achieves similar results to MTD, allowing to use different distributions in a unified manner. While D3PM (Austin et al., 2021) provides some improvements, we follow a simpler setup from MTD. We expect the improvements from D3PM to directly apply to Categorical SS-DDPM.

Table 2: Comparison of Categorical SS-DDPM and Multinomial Text Diffusion on `text8`. NLL is estimated via ELBO.

| Model   | NLL (bits/char) |
|---------|-----------------|
| MTD     | $\leq 1.72$     |
| SS-DDPM | $\leq \mathbf{1.69}$ |

**Image data**  Finally, we evaluate SS-DDPM on CIFAR-10. Since the training data is constrained to a $[0, 1]$ segment, we use Beta distributions. We evaluate our model with various numbers of generation steps, as described in equation (25), and report the resulting Fréchet Inception Distance (FID, (Heusel et al., 2017)) in Figure 6. Beta SS-DDPM achieves comparable quality with the Improved DDPM (Nichol & Dhariwal, 2021) and is slightly better on lower numbers of steps. As expected, DDIM performs better when the number of diffusion steps is low, but both SS-DDPM and DDPM outperform DDIM on longer runs. The best FID score achieved by Beta SS-DDPM is 3.17. Although the FID curves for DDPM and DDIM do not achieve this score in Figure 6, Ho et al. (2020) reported an FID score of 3.17 for 1000 DDPM steps, meaning that SS-DDPM performs similarly to DDPM in this setting.

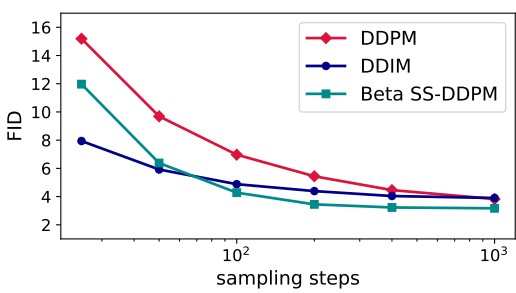

Figure 6: Quality of images, generated using Beta SS-DDPM, DDPM and DDIM with different numbers of sampling steps. Models are trained and evaluated on CIFAR-10. DDPM and DDIM results were taken from Nichol & Dhariwal (2021).

Table 3: Experiments results. The first row is real data and the second is generated samples. For the von Mises–Fisher and Dirichlet models, we show two-dimensional histograms of samples. For the Wishart model, we draw ellipses, corresponding to the p.d. matrices $x_0$ and $x_\theta$. The darker the pixel, the more ellipses pass through that pixel.

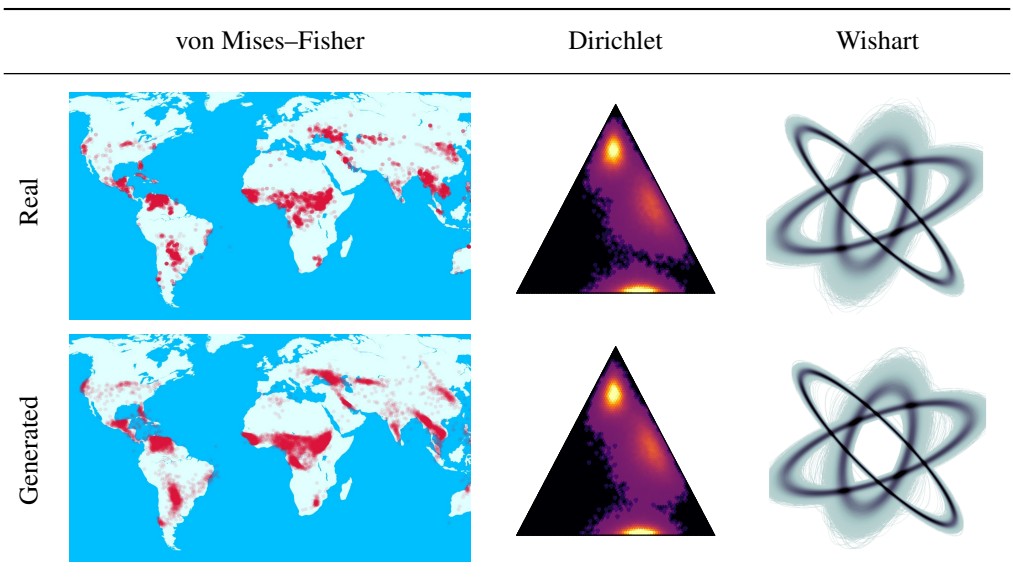

# 6   Conclusion

We propose an alternative view on diffusion-like probabilistic models. We reveal the duality between star-shaped and Markov diffusion processes that allows us to go beyond Gaussian noise by switching to a star-shaped formulation. It allows us to define diffusion-like models with arbitrary noising distributions and to establish diffusion processes on specific manifolds. We propose an efficient way to construct a reverse process for such models in the case when the noising process lies in a general subset of the exponential family and show that star-shaped diffusion models can be trained on a variety of domains with different noising distributions. On image data, star-shaped diffusion with Beta distributed noise attains comparable performance to Gaussian DDPM, challenging the optimality of Gaussian noise in this setting. The star-shaped formulation opens new applications of diffusion-like probabilistic models, especially for data from exotic domains where domain-specific non-Gaussian diffusion is more appropriate.

# Acknowledgements

We are grateful to Sergey Kholkin for additional experiments with von Mises–Fisher SS-DDPM on geodesic data and to Tingir Badmaev for helpful recommendations in experiments on image data. We'd also like to thank Viacheslav Meshchaninov for sharing his knowledge of diffusion models for text data. This research was supported in part through computational resources of HPC facilities at HSE University. The results on image data (**Image data** in Section 5; Section L) were obtained by Andrey Okhotin and Aibek Alanov with the support of the grant for research centers in the field of AI provided by the Analytical Center for the Government of the Russian Federation (ACRF) in accordance with the agreement on the provision of subsidies (identifier of the agreement 000000D730321P5Q0002) and the agreement with HSE University No. 70-2021-00139.

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

# A  Variational lower bound for the SS-DDPM model

$$\mathcal{L}^{\text{SS}}(\theta) = \mathbb{E}_{q^{\text{SS}}(x_{0:T})} \log \frac{p_\theta^{\text{SS}}(x_{0:T})}{q^{\text{SS}}(x_{1:T}|x_0)} = \mathbb{E}_{q^{\text{SS}}(x_{0:T})} \log \frac{p_\theta^{\text{SS}}(x_0|x_{1:T}) p_\theta^{\text{SS}}(x_T) \prod_{t=2}^{T} p_\theta^{\text{SS}}(x_{t-1}|x_{t:T})}{\prod_{t=1}^{T} q^{\text{SS}}(x_t|x_0)} = \tag{26}$$

$$= \mathbb{E}_{q^{\text{SS}}(x_{0:T})} \left[ \log p_\theta^{\text{SS}}(x_0|x_{1:T}) + \sum_{t=2}^{T} \log \frac{p_\theta^{\text{SS}}(x_{t-1}|x_{t:T})}{q^{\text{SS}}(x_{t-1}|x_0)} + \log \frac{p_\theta^{\text{SS}}(x_T)}{\cancel{q^{\text{SS}}(x_T|x_0)}} \right] = \tag{27}$$

$$= \mathbb{E}_{q^{\text{SS}}(x_{0:T})} \left[ \log p_\theta^{\text{SS}}(x_0|x_{1:T}) - \sum_{t=2}^{T} D_{KL}\left( q^{\text{SS}}(x_{t-1}|x_0) \,\|\, p_\theta^{\text{SS}}(x_{t-1}|x_{t:T}) \right) \right] \tag{28}$$

# B  True reverse process for Markovian and Star-Shaped DDPM

If the forward process is Markovian, the corresponding true reverse process is Markovian too:

$$q^{\text{DDPM}}(x_{t-1}|x_{t:T}) = \frac{q^{\text{DDPM}}(x_{t-1:T})}{q^{\text{DDPM}}(x_{t:T})} = \frac{q^{\text{DDPM}}(x_{t-1}) q^{\text{DDPM}}(x_t|x_{t-1}) \prod_{s=t+1}^{T} \cancel{q^{\text{DDPM}}(x_s|x_{s-1})}}{q^{\text{DDPM}}(x_t) \prod_{s=t+1}^{T} \cancel{q^{\text{DDPM}}(x_s|x_{s-1})}} = q^{\text{DDPM}}(x_{t-1}|x_t) \tag{29}$$

$$q^{\text{DDPM}}(x_{0:T}) = q(x_0) \prod_{t=1}^{T} q^{\text{DDPM}}(x_t|x_{t-1}) = q^{\text{DDPM}}(x_T) \prod_{t=1}^{T} q^{\text{DDPM}}(x_{t-1}|x_{t:T}) = q^{\text{DDPM}}(x_T) \prod_{t=1}^{T} q^{\text{DDPM}}(x_{t-1}|x_t) \tag{30}$$

For star-shaped models, however, the reverse process has a general structure that cannot be reduced further:

$$q^{\text{SS}}(x_{0:T}) = q(x_0) \prod_{t=1}^{T} q^{\text{SS}}(x_t|x_0) = q^{\text{SS}}(x_T) \prod_{t=1}^{T} q^{\text{SS}}(x_{t-1}|x_{t:T}) \tag{31}$$

In this case, a Markovian reverse process can be a very poor approximation to the true reverse process. Choosing such an approximation adds an irreducible gap to the variational lower bound:

$$\mathcal{L}^{\text{SS}}_{Markov}(\theta) = \mathbb{E}_{q^{\text{SS}}(x_{0:T})} \log \frac{p_\theta(x_T) \prod_{t=1}^{T} p_\theta(x_{t-1}|x_t)}{q^{\text{SS}}(x_{1:T}|x_0)} = \tag{32}$$

$$= \mathbb{E}_{q^{\text{SS}}(x_{0:T})} \log \frac{p_\theta(x_T) \prod_{t=1}^{T} p_\theta(x_{t-1}|x_t) q(x_0) \prod_{t=1}^{T} q^{\text{SS}}(x_{t-1}|x_t)}{q^{\text{SS}}(x_T) \prod_{t=1}^{T} q^{\text{SS}}(x_{t-1}|x_{t:T}) \prod_{t=1}^{T}} = \tag{33}$$

$$= \mathbb{E}_{q^{\text{SS}}(x_{0:T})} \left[ \log q(x_0) - \underbrace{D_{KL}\left( q^{\text{SS}}(x_T) \,\|\, p_\theta(x_T) \right) - \sum_{t=1}^{T} D_{KL}\left( q^{\text{SS}}(x_{t-1}|x_t) \,\|\, p_\theta(x_{t-1}|x_t) \right)}_{\text{Reducible}} - \tag{34}$$

$$- \underbrace{\sum_{t=1}^{T} D_{KL}\left( q^{\text{SS}}(x_{t-1}|x_{t:T}) \,\|\, q^{\text{SS}}(x_{t-1}|x_t) \right)}_{\text{Irreducible}} \right] \tag{35}$$

Intuitively, there is little information shared between $x_{t-1}$ and $x_t$, as they are conditionally independent given $x_0$. Therefore, we would expect the distribution $q^{\text{SS}}(x_{t-1}|x_t)$ to have a much higher entropy than the distribution $q^{\text{SS}}(x_{t-1}|x_{t:T})$, making the irreducible gap (35) large. The dramatic effect of this gap is illustrated in Figure 3.

This gap can also be computed analytically for Gaussian DDPMs when the data is coming from a standard Gaussian distribution $q(x_0) = \mathcal{N}(x_0; 0, 1)$. According to equations (34–35), the best Markovian reverse process in this case is $p_\theta(x_{t-1}|x_t) = q^{\text{SS}}(x_{t-1}|x_t)$. It results in the following value of the variational lower bound:

$$\mathcal{L}^{\text{SS}}_{Markov} = -\mathcal{H}[q(x_0)] + \mathcal{H}[q^{\text{SS}}(x_{0:T})] - \mathcal{H}[q^{\text{SS}}(x_T)] - \frac{1}{2} \sum_{t=1}^{T} \left[ 1 + \log(2\pi(1 - \overline{\alpha}_{t-1}^{\text{SS}} \overline{\alpha}_t^{\text{SS}})) \right] \tag{36}$$

If the reverse process is matched exactly (and has a general structure), the variational lower bound reduces to the negative entropy of the data distribution:

$$\mathcal{L}_*^{\text{SS}} = \mathbb{E}_{q^{\text{SS}}(x_{0:T})} \log \frac{p_*^{\text{SS}}(x_{0:T})}{q^{\text{SS}}(x_{1:T}|x_0)} = \tag{37}$$

$$= \mathbb{E}_{q^{\text{SS}}(x_{0:T})} \log \frac{q^{\text{SS}}(x_{0:T})}{q^{\text{SS}}(x_{1:T}|x_0)} = \tag{38}$$

$$= \mathbb{E}_{q(x_0)} \log q(x_0) = -\frac{1}{2}\log(2\pi) - \frac{1}{2} \tag{39}$$

As shown in Figure 7, the Markovian approximation adds a substantial irreducible gap to the variational lower bound.

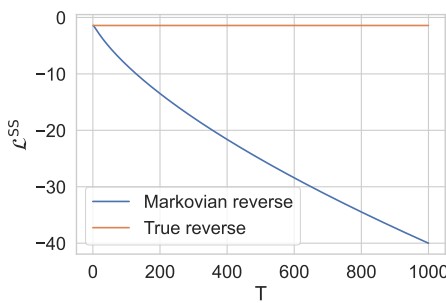

Figure 7: Variational lower bound for a Gaussian Star-Shaped DDPM model computed with the true reverse process and with the best Markovian approximation.

## C   Proof of Theorem 1

We start by establishing the following lemma. It would allow us to change the variables in the condition of a conditional distribution. Essentially, we would like to show that if a variable $y$ is only dependent on $x$ through some function $h(x)$, we can write the distribution $p(y|x)$ as $p(y|z)|_{z=h(x)}$.

**Lemma 1.** *Assume random variables $x$ and $y$ follow a joint distribution $p(x,y) = p(x)p(y|x)$, where $p(y|x)$ is given by $f(y, h(x))$. Then $p(y|x) = p(y|z)|_{z=h(x)}$, where the variable $z$ is defined as $z = h(x)$.*

*Proof.* We can write down the joint distribution over all variables as follows:

$$p(x, y, z) = p(x)f(y, h(x))\delta(z - h(x)). \tag{40}$$

By integrating $x$ out, we get

$$p(y, z) = \int p(x, y, z)dx = \int p(x)f(y, h(x))\delta(z - h(x))dx = \int p(x)f(y, z)\delta(z - h(x))dx = \tag{41}$$

$$= f(y, z)\int p(x)\delta(z - h(x))dx = f(y, z)p(z) \tag{42}$$

Finally, we obtain

$$p(y|z)|_{z=h(x)} = \left.\frac{p(y, z)}{p(z)}\right||_{z=h(x)} = f(y, z)|_{z=h(x)} = f(y, h(x)) = p(y|x) \tag{43}$$

$\square$

**Theorem 1.** *Assume the forward process of a star-shaped model takes the following form:*

$$q^{\text{SS}}(x_t|x_0) = h_t(x_t)\exp\left\{\eta_t(x_0)^\mathsf{T}\mathcal{T}(x_t) - \Omega_t(x_0)\right\}, \tag{13}$$
$$\eta_t(x_0) = A_t f(x_0) + b_t. \tag{14}$$

*Let $G_t$ be a tail statistic, defined as follows:*

$$G_t = \mathcal{G}_t(x_{t:T}) = \sum_{s=t}^{T} A_s^\mathsf{T}\mathcal{T}(x_s). \tag{15}$$

*Then, $G_t$ is a sufficient tail statistic:*

$$q^{\text{SS}}(x_{t-1}|x_{t:T}) = q^{\text{SS}}(x_{t-1}|G_t). \tag{16}$$

*Proof.*

$$q^{\text{SS}}(x_{t-1}|x_{t:T}) = \int q^{\text{SS}}(x_{t-1}|x_0)q^{\text{SS}}(x_0|x_{t:T})dx_0 \tag{44}$$

$$q^{\text{SS}}(x_0|x_{t:T}) = \frac{q(x_0)\prod_{s=t}^{T}q^{\text{SS}}(x_s|x_0)}{q^{\text{SS}}(x_{t:T})} = \frac{q(x_0)}{q^{\text{SS}}(x_{t:T})}\left(\prod_{s=t}^{T}h_s(x_s)\right)\exp\left\{\sum_{s=t}^{T}\left(\eta_s(x_0)^\mathsf{T}\mathcal{T}(x_s) - \Omega_s(x_0)\right)\right\} = \tag{45}$$

$$= \frac{q(x_0)}{q^{\text{SS}}(x_{t:T})} \left( \prod_{s=t}^T h_s(x_s) \right) \exp \left\{ \sum_{s=t}^T \left( (A_s f(x_0) + b_s)^\mathsf{T} \mathcal{T}(x_s) - \Omega_s(x_0) \right) \right\} = \tag{46}$$

$$= \frac{q(x_0)}{q^{\text{SS}}(x_{t:T})} \left( \prod_{s=t}^T h_s(x_s) \right) \exp \left\{ f(x_0)^\mathsf{T} \sum_{s=t}^T A_s^\mathsf{T} \mathcal{T}(x_s) + \sum_{s=t}^T \left( b_s^\mathsf{T} \mathcal{T}(x_s) - \Omega_s(x_0) \right) \right\} = \tag{47}$$

$$= \frac{q(x_0)}{q^{\text{SS}}(x_{t:T})} \left( \prod_{s=t}^T h_s(x_s) \right) \exp \left\{ f(x_0)^\mathsf{T} G_t + \sum_{s=t}^T \left( b_s^\mathsf{T} \mathcal{T}(x_s) - \Omega_s(x_0) \right) \right\} = \begin{bmatrix} \text{a distribution must} \\ \text{be normalized} \end{bmatrix} = \tag{48}$$

$$= \frac{q(x_0) \exp \left\{ f(x_0)^\mathsf{T} G_t - \sum_{s=t}^T \Omega_s(x_0) \right\}}{\int q(x_0) \exp \left\{ f(x_0)^\mathsf{T} G_t - \sum_{s=t}^T \Omega_s(x_0) \right\} dx_0} = \begin{bmatrix} \text{Lemma 1 for } G_t \text{ as } z, \\ x_0 \text{ as } y \text{ and } x_{t:T} \text{ as } x \end{bmatrix} = q^{\text{SS}}(x_0|G_t) \tag{49}$$

$$q^{\text{SS}}(x_{t-1}|x_{t:T}) = \int q^{\text{SS}}(x_{t-1}|x_0) q^{\text{SS}}(x_0|x_{t:T}) dx_0 = \int q^{\text{SS}}(x_{t-1}|x_0) q^{\text{SS}}(x_0|G_t) dx_0 = q^{\text{SS}}(x_{t-1}|G_t) \tag{50}$$

$$\square$$

# D  Gaussian SS-DDPM is equivalent to Gaussian DDPM

**Theorem 2.** *Let $\overline{\alpha}_t^{\text{DDPM}}$ define the noising schedule for a DDPM model (1–2) via $\beta_t = (\overline{\alpha}_{t-1}^{\text{DDPM}} - \overline{\alpha}_t^{\text{DDPM}})/\overline{\alpha}_{t-1}^{\text{DDPM}}$. Let $q^{\text{SS}}(x_{0:T})$ be a Gaussian SS-DDPM forward process with the following noising schedule and sufficient tail statistic:*

$$q^{\text{SS}}(x_t|x_0) = \mathcal{N} \left( x_t; \sqrt{\overline{\alpha}_t^{\text{SS}}} x_0, 1 - \overline{\alpha}_t^{\text{SS}} \right), \tag{22}$$

$$\mathcal{G}_t(x_{t:T}) = \frac{1 - \overline{\alpha}_t^{\text{DDPM}}}{\sqrt{\overline{\alpha}_t^{\text{DDPM}}}} \sum_{s=t}^T \frac{\sqrt{\overline{\alpha}_s^{\text{SS}}} x_s}{1 - \overline{\alpha}_s^{\text{SS}}}, \text{ where} \tag{23}$$

$$\frac{\overline{\alpha}_t^{\text{SS}}}{1 - \overline{\alpha}_t^{\text{SS}}} = \frac{\overline{\alpha}_t^{\text{DDPM}}}{1 - \overline{\alpha}_t^{\text{DDPM}}} - \frac{\overline{\alpha}_{t+1}^{\text{DDPM}}}{1 - \overline{\alpha}_{t+1}^{\text{DDPM}}}. \tag{24}$$

*Then the tail statistic $G_t$ follows a Gaussian DDPM noising process $q^{\text{DDPM}}(x_{0:T})|_{x_{1:T}=G_{1:T}}$ defined by the schedule $\overline{\alpha}_t^{\text{DDPM}}$. Moreover, the corresponding reverse processes and VLB objectives are also equivalent.*

*Proof.* We start by listing the necessary definitions and expressions used in the Gaussian DDPM model (Ho et al., 2020).

$$q^{\text{DDPM}}(x_{0:T}) = q(x_0) \prod_{i=1}^T q^{\text{DDPM}}(x_t|x_{t-1}) \tag{51}$$

$$q^{\text{DDPM}}(x_t|x_{t-1}) = \mathcal{N} \left( x_t; \sqrt{1 - \beta_t} x_{t-1}, \beta_t \mathbf{I} \right) \tag{52}$$

$$\alpha_s^{\text{DDPM}} = 1 - \beta_s^{\text{DDPM}} \tag{53}$$

$$\overline{\alpha}_t^{\text{DDPM}} = \prod_{s=1}^t \alpha_s^{\text{DDPM}} \tag{54}$$

$$q^{\text{DDPM}}(x_t|x_0) = \mathcal{N} \left( x_t; \sqrt{\overline{\alpha}_t^{\text{DDPM}}} x_0, (1 - \overline{\alpha}_t^{\text{DDPM}}) \mathbf{I} \right) \tag{55}$$

$$\tilde{\mu}_t(x_t, x_0) = \frac{\sqrt{\overline{\alpha}_{t-1}^{\text{DDPM}}} \beta_t}{1 - \overline{\alpha}_t^{\text{DDPM}}} x_0 + \frac{\sqrt{\alpha_t^{\text{DDPM}}}(1 - \overline{\alpha}_{t-1}^{\text{DDPM}})}{1 - \overline{\alpha}_t^{\text{DDPM}}} x_t \tag{56}$$

$$\tilde{\beta}_t^{\text{DDPM}} = \frac{1 - \overline{\alpha}_{t-1}^{\text{DDPM}}}{1 - \overline{\alpha}_t^{\text{DDPM}}} \beta_t \tag{57}$$

$$q^{\text{DDPM}}(x_{t-1}|x_t, x_0) = \mathcal{N} \left( x_{t-1}; \tilde{\mu}_t(x_t, x_0), \tilde{\beta}_t^{\text{DDPM}} \mathbf{I} \right) \tag{58}$$

$$p_\theta^{\mathrm{DDPM}}(x_{0:T}) = q^{\mathrm{DDPM}}(x_T) \prod_{i=1}^{T} p_\theta^{\mathrm{DDPM}}(x_{t-1}|x_t) \tag{59}$$

$$p_\theta^{\mathrm{DDPM}}(x_{t-1}|x_t) = \mathcal{N}\left(x_{t-1}; \tilde{\mu}_t(x_t, x_\theta^{\mathrm{DDPM}}(x_t, t)), \tilde{\beta}_t^{\mathrm{DDPM}}\mathbf{I}\right) \tag{60}$$

In this notation, the variational lower bound for DDPM can be written as follows (here we omit the term corresponding to $x_T$ because it's either zero or a small constant):

$$\mathcal{L}^{\mathrm{DDPM}}(\theta) = \mathbb{E}_{q^{\mathrm{DDPM}}(x_{0:T})}\left[\log p_\theta^{\mathrm{DDPM}}(x_0|x_1) - \sum_{t=2}^{T} D_{KL}\left(q^{\mathrm{DDPM}}(x_{t-1}|x_t, x_0) \,\|\, p_\theta^{\mathrm{DDPM}}(x_{t-1}|x_t)\right)\right] = \tag{61}$$

$$= \mathbb{E}_{q^{\mathrm{DDPM}}(x_{0:T})}\left[\log p_\theta^{\mathrm{DDPM}}(x_0|x_1) - \sum_{t=2}^{T} D_{KL}\left(q^{\mathrm{DDPM}}(x_{t-1}|x_t, x_0) \,\|\, q^{\mathrm{DDPM}}(x_{t-1}|x_t, x_0)|_{x_0 = x_\theta^{\mathrm{DDPM}}(x_t, t)}\right)\right] \tag{62}$$

$$D_{KL}\left(q^{\mathrm{DDPM}}(x_{t-1}|x_t, x_0) \,\|\, q^{\mathrm{DDPM}}(x_{t-1}|x_t, x_0)|_{x_0 = x_\theta^{\mathrm{DDPM}}(x_t, t)}\right) = \frac{\left(\frac{\sqrt{\overline{\alpha}_{t-1}^{\mathrm{DDPM}}}\beta_t^{\mathrm{DDPM}}}{1-\overline{\alpha}_t^{\mathrm{DDPM}}}(x_0 - x_\theta^{\mathrm{DDPM}}(x_t, t))\right)^2}{2\tilde{\beta}_t^{\mathrm{DDPM}}} \tag{63}$$

For convenience, we additionally define $\overline{\alpha}_{T+1}^{\mathrm{DDPM}} = 0$. Since all the involved distributions are typically isotropic, we consider a one-dimensional case without the loss of generality.

First, we show that the forward processes $q^{\mathrm{SS}}(x_0, G_{1:T})$ and $q^{\mathrm{DDPM}}(x_{0:T})$ are equivalent.

$$G_t = \frac{1 - \overline{\alpha}_t^{\mathrm{DDPM}}}{\sqrt{\overline{\alpha}_t^{\mathrm{DDPM}}}} \sum_{s=t}^{T} \frac{\sqrt{\overline{\alpha}_s^{\mathrm{SS}}} x_s}{1 - \overline{\alpha}_s^{\mathrm{SS}}} \tag{64}$$

Since $G_t$ is a linear combination of independent (given $x_0$) Gaussian random variables, its distribution (given $x_0$) is also Gaussian. All the terms conveniently cancel out, and we recover the same form as the forward process of DDPM.

$$q^{\mathrm{SS}}(G_t|x_0) = \mathcal{N}\left(G_t; \frac{1 - \overline{\alpha}_t^{\mathrm{DDPM}}}{\sqrt{\overline{\alpha}_t^{\mathrm{DDPM}}}} \sum_{s=t}^{T} \frac{\overline{\alpha}_s^{\mathrm{SS}} x_0}{1 - \overline{\alpha}_s^{\mathrm{SS}}}; \left(\frac{1 - \overline{\alpha}_t^{\mathrm{DDPM}}}{\sqrt{\overline{\alpha}_t^{\mathrm{DDPM}}}}\right)^2 \sum_{s=t}^{T}\left(\frac{\overline{\alpha}_s^{\mathrm{SS}}}{(1 - \overline{\alpha}_s^{\mathrm{SS}})^2}(1 - \overline{\alpha}_s^{\mathrm{SS}})\right)\right) = \tag{65}$$

$$= \mathcal{N}\left(G_t; \frac{1 - \overline{\alpha}_t^{\mathrm{DDPM}}}{\sqrt{\overline{\alpha}_t^{\mathrm{DDPM}}}} \frac{\overline{\alpha}_t^{\mathrm{DDPM}}}{1 - \overline{\alpha}_t^{\mathrm{DDPM}}} x_0; \left(\frac{1 - \overline{\alpha}_t^{\mathrm{DDPM}}}{\sqrt{\overline{\alpha}_t^{\mathrm{DDPM}}}}\right)^2 \frac{\overline{\alpha}_t^{\mathrm{DDPM}}}{1 - \overline{\alpha}_t^{\mathrm{DDPM}}}\right) = \tag{66}$$

$$= \mathcal{N}\left(G_t; \sqrt{\overline{\alpha}_t^{\mathrm{DDPM}}} x_0; (1 - \overline{\alpha}_t^{\mathrm{DDPM}})\right) = q^{\mathrm{DDPM}}(x_t|x_0)|_{x_t = G_t} \tag{67}$$

Since both processes share the same $q(x_0)$, and have the same Markovian structure, matching these marginals sufficiently shows that the processes are equivalent:

$$q^{\mathrm{SS}}(x_0, G_{1:T}) = q^{\mathrm{DDPM}}(x_{0:T})|_{x_{1:T} = G_{1:T}} \tag{68}$$

Consequently, the posteriors are matching too:

$$q^{\mathrm{SS}}(G_{t-1}|G_t, x_0) = q^{\mathrm{DDPM}}(x_{t-1}|x_t, x_0)|_{x_{t-1,t} = G_{t-1,t}} \tag{69}$$

Next, we show that the reverse processes $p_\theta^{\mathrm{SS}}(G_{t-1}|G_t)$ and $p_\theta^{\mathrm{DDPM}}(x_{t-1}|x_t)$ are the same. Since $G_t$ in SS-DDPM is the same as $x_t$ in DDPM, we can assume the predictive models $x_\theta^{\mathrm{SS}}(G_t, t)$ and $x_\theta^{\mathrm{DDPM}}(x_t, t)$ to coincide at $x_t = G_t$.

$$G_{t-1} = \frac{1 - \overline{\alpha}_{t-1}^{\mathrm{DDPM}}}{\sqrt{\overline{\alpha}_{t-1}^{\mathrm{DDPM}}}} \sum_{s=t-1}^{T} \frac{\sqrt{\overline{\alpha}_s^{\mathrm{SS}}} x_s}{1 - \overline{\alpha}_s^{\mathrm{SS}}} = \frac{1 - \overline{\alpha}_{t-1}^{\mathrm{DDPM}}}{\sqrt{\overline{\alpha}_{t-1}^{\mathrm{DDPM}}}} \frac{\sqrt{\overline{\alpha}_{t-1}^{\mathrm{SS}}} x_{t-1}}{1 - \overline{\alpha}_{t-1}^{\mathrm{SS}}} + \frac{1 - \overline{\alpha}_{t-1}^{\mathrm{DDPM}}}{\sqrt{\overline{\alpha}_{t-1}^{\mathrm{DDPM}}}} \frac{\sqrt{\overline{\alpha}_t^{\mathrm{DDPM}}}}{1 - \overline{\alpha}_t^{\mathrm{DDPM}}} G_t = \tag{70}$$

$$= \frac{1 - \overline{\alpha}_{t-1}^{\mathrm{DDPM}}}{\sqrt{\overline{\alpha}_{t-1}^{\mathrm{DDPM}}}} \sum_{s=t-1}^{T} \frac{\sqrt{\overline{\alpha}_s^{\mathrm{SS}}} x_s}{1 - \overline{\alpha}_s^{\mathrm{SS}}} = \frac{1 - \overline{\alpha}_{t-1}^{\mathrm{DDPM}}}{\sqrt{\overline{\alpha}_{t-1}^{\mathrm{DDPM}}}} \frac{\sqrt{\overline{\alpha}_{t-1}^{\mathrm{SS}}} x_{t-1}}{1 - \overline{\alpha}_{t-1}^{\mathrm{SS}}} + \frac{\sqrt{\overline{\alpha}_t^{\mathrm{DDPM}}}(1 - \overline{\alpha}_{t-1}^{\mathrm{DDPM}})}{1 - \overline{\alpha}_t^{\mathrm{DDPM}}} G_t, \tag{71}$$

where $x_{t-1} \sim p_\theta^{\mathrm{SS}}(x_{t-1}|G_t) = q^{\mathrm{SS}}(x_{t-1}|x_0)|_{x_0 = x_\theta(G_t, t)}$. Therefore, $p_\theta^{\mathrm{SS}}(G_{t-1}|G_t)$ is also a Gaussian distribution. Let's take care of the mean first:

$$\mathbb{E}_{p_\theta^{\mathrm{SS}}(G_{t-1}|G_t)} G_{t-1} = \frac{1 - \overline{\alpha}_{t-1}^{\mathrm{DDPM}}}{\sqrt{\overline{\alpha}_{t-1}^{\mathrm{DDPM}}}} \frac{\overline{\alpha}_{t-1}^{\mathrm{SS}}}{1 - \overline{\alpha}_{t-1}^{\mathrm{SS}}} x_\theta(G_t, t) + \frac{\sqrt{\overline{\alpha}_t^{\mathrm{DDPM}}}(1 - \overline{\alpha}_{t-1}^{\mathrm{DDPM}})}{1 - \overline{\alpha}_t^{\mathrm{DDPM}}} G_t = \tag{72}$$

$$= \frac{1-\overline{\alpha}_{t-1}^{\text{DDPM}}}{\sqrt{\overline{\alpha}_{t-1}^{\text{DDPM}}}}\left(\frac{\overline{\alpha}_{t-1}^{\text{DDPM}}}{1-\overline{\alpha}_{t-1}^{\text{DDPM}}} - \frac{\overline{\alpha}_t^{\text{DDPM}}}{1-\overline{\alpha}_t^{\text{DDPM}}}\right) x_\theta(G_t, t) + \frac{\sqrt{\alpha_t^{\text{DDPM}}}(1-\overline{\alpha}_{t-1}^{\text{DDPM}})}{1-\overline{\alpha}_t^{\text{DDPM}}} G_t = \tag{73}$$

$$= \left(\sqrt{\overline{\alpha}_{t-1}^{\text{DDPM}}} - \frac{(1-\overline{\alpha}_{t-1}^{\text{DDPM}})\sqrt{\overline{\alpha}_{t-1}^{\text{DDPM}}}\alpha_t^{\text{DDPM}}}{1-\overline{\alpha}_t^{\text{DDPM}}}\right) x_\theta(G_t, t) + \frac{\sqrt{\alpha_t^{\text{DDPM}}}(1-\overline{\alpha}_{t-1}^{\text{DDPM}})}{1-\overline{\alpha}_t^{\text{DDPM}}} G_t = \tag{74}$$

$$= \left(\frac{1-\overline{\alpha}_t^{\text{DDPM}} - (1-\overline{\alpha}_{t-1}^{\text{DDPM}})\alpha_t^{\text{DDPM}}}{1-\overline{\alpha}_t^{\text{DDPM}}}\right)\sqrt{\overline{\alpha}_{t-1}^{\text{DDPM}}}x_\theta(G_t, t) + \frac{\sqrt{\alpha_t^{\text{DDPM}}}(1-\overline{\alpha}_{t-1}^{\text{DDPM}})}{1-\overline{\alpha}_t^{\text{DDPM}}} G_t = \tag{75}$$

$$= \left(\frac{1-\overline{\alpha}_t^{\text{DDPM}} - \alpha_t^{\text{DDPM}} + \overline{\alpha}_t^{\text{DDPM}}}{1-\overline{\alpha}_t^{\text{DDPM}}}\right)\sqrt{\overline{\alpha}_{t-1}^{\text{DDPM}}}x_\theta(G_t, t) + \frac{\sqrt{\alpha_t^{\text{DDPM}}}(1-\overline{\alpha}_{t-1}^{\text{DDPM}})}{1-\overline{\alpha}_t^{\text{DDPM}}} G_t = \tag{76}$$

$$= \frac{\sqrt{\overline{\alpha}_{t-1}^{\text{DDPM}}}\beta_t}{1-\overline{\alpha}_t^{\text{DDPM}}} x_\theta(G_t, t) + \frac{\sqrt{\alpha_t^{\text{DDPM}}}(1-\overline{\alpha}_{t-1}^{\text{DDPM}})}{1-\overline{\alpha}_t^{\text{DDPM}}} G_t \tag{77}$$

Now, let's derive the variance:

$$\mathbb{D}_{p_\theta^{\text{SS}}(G_{t-1}|G_t)}G_{t-1} = \left(\frac{1-\overline{\alpha}_{t-1}^{\text{DDPM}}}{\sqrt{\overline{\alpha}_{t-1}^{\text{DDPM}}}} \frac{\sqrt{\overline{\alpha}_{t-1}^{\text{SS}}}}{1-\overline{\alpha}_{t-1}^{\text{SS}}}\right)^2 (1-\overline{\alpha}_{t-1}^{\text{SS}}) = \tag{78}$$

$$= \left(\frac{1-\overline{\alpha}_{t-1}^{\text{DDPM}}}{\sqrt{\overline{\alpha}_{t-1}^{\text{DDPM}}}} \sqrt{\frac{\overline{\alpha}_{t-1}^{\text{SS}}}{1-\overline{\alpha}_{t-1}^{\text{SS}}}}\right)^2 = \frac{(1-\overline{\alpha}_{t-1}^{\text{DDPM}})^2}{\overline{\alpha}_{t-1}^{\text{DDPM}}}\frac{\overline{\alpha}_{t-1}^{\text{SS}}}{1-\overline{\alpha}_{t-1}^{\text{SS}}} = \frac{(1-\overline{\alpha}_{t-1}^{\text{DDPM}})^2}{\overline{\alpha}_{t-1}^{\text{DDPM}}}\left(\frac{\overline{\alpha}_{t-1}^{\text{DDPM}}}{1-\overline{\alpha}_{t-1}^{\text{DDPM}}} - \frac{\overline{\alpha}_t^{\text{DDPM}}}{1-\overline{\alpha}_t^{\text{DDPM}}}\right) = \tag{79}$$

$$= (1-\overline{\alpha}_{t-1}^{\text{DDPM}})\left(1 - \frac{(1-\overline{\alpha}_{t-1}^{\text{DDPM}})\alpha_t^{\text{DDPM}}}{1-\overline{\alpha}_t^{\text{DDPM}}}\right) = \frac{(1-\overline{\alpha}_{t-1}^{\text{DDPM}})(1-\overline{\alpha}_t^{\text{DDPM}} - (1-\overline{\alpha}_{t-1}^{\text{DDPM}})\alpha_t^{\text{DDPM}})}{1-\overline{\alpha}_t^{\text{DDPM}}} = \tag{80}$$

$$= \frac{1-\overline{\alpha}_{t-1}^{\text{DDPM}}}{1-\overline{\alpha}_t^{\text{DDPM}}}(1-\overline{\alpha}_t^{\text{DDPM}} - \alpha_t^{\text{DDPM}} + \overline{\alpha}_t^{\text{DDPM}}) = \frac{1-\overline{\alpha}_{t-1}^{\text{DDPM}}}{1-\overline{\alpha}_t^{\text{DDPM}}}\beta_t^{\text{DDPM}} = \tilde{\beta}_t^{\text{DDPM}} \tag{81}$$

Therefore,

$$p_\theta^{\text{SS}}(G_{t-1}|G_t) = \mathcal{N}\left(G_{t-1}; \frac{\sqrt{\overline{\alpha}_{t-1}^{\text{DDPM}}}\beta_t}{1-\overline{\alpha}_t^{\text{DDPM}}}x_\theta(G_t, t) + \frac{\sqrt{\alpha_t^{\text{DDPM}}}(1-\overline{\alpha}_{t-1}^{\text{DDPM}})}{1-\overline{\alpha}_t^{\text{DDPM}}}G_t, \tilde{\beta}_t^{\text{DDPM}}\right) = p_\theta^{\text{DDPM}}(x_{t-1}|x_t)\big|_{x_{t-1,t}=G_{t-1,t}} \tag{82}$$

Finally, we show that the variational lower bounds are the same.

$$\mathcal{L}^{\text{SS}}(\theta) = \mathbb{E}_{q^{\text{SS}}(x_{0:T})}\left[\log p_\theta^{\text{SS}}(x_0|G_1) - \sum_{t=2}^T D_{KL}\left(q^{\text{SS}}(x_{t-1}|x_0) \,\|\, p_\theta^{\text{SS}}(x_{t-1}|G_t)\right)\right] = \tag{83}$$

$$= \mathbb{E}_{q^{\text{SS}}(x_{0:T})}\left[\log p_\theta^{\text{SS}}(x_0|G_1) - \sum_{t=2}^T D_{KL}\left(q^{\text{SS}}(x_{t-1}|x_0) \,\Big\|\, q^{\text{SS}}(x_{t-1}|x_0)\big|_{x_0=x_\theta(G_t,t)}\right)\right] \tag{84}$$

Since $G_1$ from the star-shaped model is the same as $x_1$ from the DDPM model, the first term $\log p_\theta^{\text{SS}}(x_0|G_1)$ coincides with $\log p_\theta^{\text{DDPM}}(x_0|x_1)\big|_{x_1=G_1}$.

$$D_{KL}\left(q^{\text{SS}}(x_{t-1}|x_0) \,\Big\|\, q^{\text{SS}}(x_{t-1}|x_0)\big|_{x_0=x_\theta(G_t,t)}\right) = \frac{\left(\sqrt{\overline{\alpha}_{t-1}^{\text{SS}}}(x_0 - x_\theta(G_t,t))\right)^2}{2(1-\overline{\alpha}_{t-1}^{\text{SS}})} = \tag{85}$$

$$= \frac{\overline{\alpha}_{t-1}^{\text{SS}}}{2(1-\overline{\alpha}_{t-1}^{\text{SS}})}(x_0 - x_\theta(G_t,t))^2 = \frac{1}{2}\left(\frac{\overline{\alpha}_{t-1}^{\text{DDPM}}}{1-\overline{\alpha}_{t-1}^{\text{DDPM}}} - \frac{\overline{\alpha}_t^{\text{DDPM}}}{1-\overline{\alpha}_t^{\text{DDPM}}}\right)(x_0 - x_\theta(G_t,t))^2 = \tag{86}$$

$$= \frac{1}{2}\frac{\overline{\alpha}_{t-1}^{\text{DDPM}} - \overline{\alpha}_t^{\text{DDPM}}}{(1-\overline{\alpha}_{t-1}^{\text{DDPM}})(1-\overline{\alpha}_t^{\text{DDPM}})}(x_0 - x_\theta(G_t,t))^2 = \frac{1}{2}\frac{\overline{\alpha}_{t-1}^{\text{DDPM}}\beta_t^{\text{DDPM}}}{(1-\overline{\alpha}_{t-1}^{\text{DDPM}})(1-\overline{\alpha}_t^{\text{DDPM}})}(x_0 - x_\theta(G_t,t))^2 = \tag{87}$$

$$= \frac{\overline{\alpha}_{t-1}^{\text{DDPM}}\beta_t^{\text{DDPM}}(x_0 - x_\theta^{\text{DDPM}}(x_t,t))^2}{2(1-\overline{\alpha}_{t-1}^{\text{DDPM}})(1-\overline{\alpha}_t^{\text{DDPM}})} = \frac{\frac{\overline{\alpha}_{t-1}^{\text{DDPM}}(\beta_t^{\text{DDPM}})^2}{(1-\overline{\alpha}_t^{\text{DDPM}})^2}(x_0 - x_\theta^{\text{DDPM}}(x_t,t))^2}{2\frac{1-\overline{\alpha}_{t-1}^{\text{DDPM}}}{1-\overline{\alpha}_t^{\text{DDPM}}}\beta_t^{\text{DDPM}}} = \tag{88}$$

$$= \frac{\left(\frac{\sqrt{\overline{\alpha}_{t-1}^{\text{DDPM}}}\beta_t^{\text{DDPM}}}{1-\overline{\alpha}_t^{\text{DDPM}}}(x_0 - x_\theta^{\text{DDPM}}(x_t,t))\right)^2}{2\tilde{\beta}_t^{\text{DDPM}}} = D_{KL}\left(q^{\text{DDPM}}(x_{t-1}|x_t,x_0) \,\Big\|\, q^{\text{DDPM}}(x_{t-1}|x_t,x_0)\big|_{x_0=x_\theta^{\text{DDPM}}(x_t,t)}\right) \tag{89}$$

Therefore, we finally obtain $\mathcal{L}^{\text{SS}}(\theta) = \mathcal{L}^{\text{DDPM}}(\theta)$. $\qquad\square$

# E Duality between star-shaped and Markovian diffusion: general case

We assume that $\mathcal{T}(x_t)$ and all matrices $A_t$ (except possibly $A_T$) are invertible. Then, there is a bijection between the set of tail statistics $G_{t:T}$ and the tail $x_{t:T}$.

First, we show that the variables $G_{1:T}$ form a Markov chain.

$$q(G_{t-1}|G_{t:T}) = q(G_{t-1}|x_{t:T}) = \int q(G_{t-1}|x_0, x_{t:T})q(x_0|x_{t:T})dx_0 = \tag{90}$$

$$= \int q(G_{t-1}|G_t, x_0)q(x_0|G_t)dx_0 = q(G_{t-1}|G_t) \tag{91}$$

$$q(x_0, G_{t:T}) = q(x_0|G_{1:T}) \prod_{t=2}^{T} q(G_{t-1}|G_{t:T}) = q(x_0|G_1) \prod_{t=2}^{T} q(G_{t-1}|G_t) = q(x_0)q(G_1|x_0) \prod_{t=2}^{T} q(G_t|G_{t-1}) \tag{92}$$

The last equation holds since the reverse of a Markov chain is also a Markov chain.

This means that a star-shaped diffusion process on $x_{1:T}$ implicitly defines some Markovian diffusion process on the tail statistics $G_{1:T}$. Due to the definition of $G_{t-1} = G_t + A_{t-1}^{\mathsf{T}}\mathcal{T}(x_{t-1})$, we can write down the following factorization of that process:

$$q(x_0, G_{t:T}) = q(x_0)q(G_T) \prod_{t=2}^{T} q(G_{t-1}|G_t, x_0) \tag{93}$$

The posteriors $q(G_{t-1}|G_t, x_0)$ can then be computed by a change of variables:

$$q(G_{t-1}|G_t, x_0) = q(x_{t-1}|x_0)\big|_{x_{t-1}=\mathcal{T}^{-1}\left(A_{t-1}^{-\mathsf{T}}(G_{t-1}-G_t)\right)} \cdot \left|\det\left[\frac{d}{dG_{t-1}}\mathcal{T}^{-1}\left(A_{t-1}^{-\mathsf{T}}(G_{t-1}-G_t)\right)\right]\right| \tag{94}$$

This also allows us to define the reverse model like it was defined in DDPM:

$$p_\theta(G_{t-1}|G_t) = q(G_{t-1}|G_t, x_0)\big|_{x_0=x_\theta(G_t, t)} \tag{95}$$

This definition is consistent with the reverse process of SS-DDPM $p_\theta(x_{t-1}|x_{t:T}) = q(x_{t-1}|x_0)\big|_{x_0=x_\theta(G_t, t)}$. Now that both the forward and reverse processes are defined, we can write down the corresponding variational lower bound. Because the model is structured exactly like a DDPM, the variational lower bound is going to look the same. However, we can show that it is equivalent to the variational lower bound of SS-DDPM:

$$\mathcal{L}^{\text{Dual}}(\theta) = \mathbb{E}_{q(x_0, G_{1:T})}\left[\log p_\theta(x_0|G_1) - \sum_{t=2}^{T} D_{KL}\left(q(G_{t-1}|G_t, x_0) \,\|\, p_\theta(G_{t-1}|G_t)\right) - D_{KL}\left(q(G_T|x_0) \,\|\, p_\theta(G_T)\right)\right] = \tag{96}$$

$$= \mathbb{E}_{q(x_0, G_{1:T})}\left[\log p_\theta(x_0|G_1) - \sum_{t=2}^{T} \mathbb{E}_{q(G_{t-1}|G_t, x_0)} \log \frac{q(G_{t-1}|G_t, x_0)}{p_\theta(G_{t-1}|G_t)} - D_{KL}\left(q(x_T|x_0) \,\|\, p_\theta(x_T)\right)\right] = \tag{97}$$

$$= \mathbb{E}_{q(x_{0:T})}\left[\log p_\theta(x_0|G_1) - \sum_{t=2}^{T} \mathbb{E}_{q(x_{t-1}|x_0)} \log \frac{q(x_{t-1}|x_0)}{p_\theta(x_{t-1}|G_t)} - D_{KL}\left(q(x_T|x_0) \,\|\, p_\theta(x_T)\right)\right] = \tag{98}$$

$$= \mathbb{E}_{q(x_{0:T})}\left[\log p_\theta(x_0|G_1) - \sum_{t=2}^{T} D_{KL}\left(q(x_{t-1}|x_0) \,\|\, p_\theta(x_{t-1}|G_t)\right) - D_{KL}\left(q(x_T|x_0) \,\|\, p_\theta(x_T)\right)\right] = \mathcal{L}^{\text{SS}}(\theta) \tag{99}$$

As we can see, there are two equivalent ways to write down the model. One is SS-DDPM, a star-shaped diffusion model, where we only need to define the marginal transition probabilities $q(x_t|x_0)$. Another way is to rewrite the model in terms of the tail statistics $G_{t:T}$. This way we obtain a non-Gaussian DDPM that is implicitly defined by the star-shaped model. Because of this equivalence, we see the star-shaped diffusion of $x_{1:T}$ and the Markovian diffusion of $G_{1:T}$ as a pair of dual processes.

# F    SS-DDPM in different families

The main principles for designing a SS-DDPM model are similar to designing a DDPM model. When $t$ goes to zero, we wish to recover the Dirac delta, centered at $x_0$:

$$q^{\text{SS}}(x_t|x_0) \xrightarrow[t \to 0]{} \delta(x_t - x_0) \tag{100}$$

When $t$ goes to $T$, we wish to obtain some standard distribution that doesn't depend on $x_0$:

$$q^{\text{SS}}(x_t|x_0) \xrightarrow[t \to T]{} q_T^{\text{SS}}(x_T) \tag{101}$$

In exponential families with linear parameterization of the natural parameter $\eta_t(x_0) = a_t f(x_0) + b_t$, we can define the schedule by choosing the parameters $a_t$ and $b_t$ that satisfy conditions (100–101). After that, we can use Theorem 1 to define the tail statistic $\mathcal{G}_t(x_{t:T})$ using the sufficient statistic $\mathcal{T}(x_t)$ of the corresponding family. However, as shown in the following sections, in some cases linear parameterization admits a simpler sufficient tail statistic.

The following sections contain examples of defining the SS-DDPM model for different families. These results are summarized in Table 6.

## F.1    Gaussian

$$q^{\text{SS}}(x_t|x_0) = \mathcal{N}\left(x_t; \sqrt{\overline{\alpha}_t^{\text{SS}}} x_0, (1 - \overline{\alpha}_t^{\text{SS}})I\right) \tag{102}$$

Since Gaussian SS-DDPM is equivalent to a Markovian DDPM, it is natural to directly reuse the schedule from a Markovian DDPM. As we show in Theorem 2, given a Markovian DDPM defined by $\overline{\alpha}_t^{\text{DDPM}}$, the following parameterization will produce the same process in the space of tail statistics:

$$\frac{\overline{\alpha}_t^{\text{SS}}}{1 - \overline{\alpha}_t^{\text{SS}}} = \frac{\overline{\alpha}_t^{\text{DDPM}}}{1 - \overline{\alpha}_t^{\text{DDPM}}} - \frac{\overline{\alpha}_{t+1}^{\text{DDPM}}}{1 - \overline{\alpha}_{t+1}^{\text{DDPM}}} \tag{103}$$

$$\mathcal{G}_t(x_{t:T}) = \frac{1 - \overline{\alpha}_t^{\text{DDPM}}}{\sqrt{\overline{\alpha}_t^{\text{DDPM}}}} \sum_{s=t}^{T} \frac{\sqrt{\overline{\alpha}_s^{\text{SS}}} x_s}{1 - \overline{\alpha}_s^{\text{SS}}} \tag{104}$$

The KL divergence is computed as follows:

$$D_{KL}\left(q^{\text{SS}}(x_t|x_0) \,\|\, p_\theta^{\text{SS}}(x_t|G_t)\right) = \frac{\overline{\alpha}_t^{\text{SS}}(x_0 - x_\theta)^2}{2(1 - \overline{\alpha}_t^{\text{SS}})} \tag{105}$$

## F.2    Beta

$$q(x_t|x_0) = \text{Beta}(x_t; \alpha_t, \beta_t) \tag{106}$$

There are many ways to define a noising schedule. We choose to fix the mode of the distribution at $x_0$ and introduce a concentration parameter $\nu_t$, parameterizing $\alpha_t = 1 + \nu_t x_0$ and $\beta_t = 1 + \nu_t(1 - x_0)$. By setting $\nu_t$ to zero, we recover a uniform distribution, and by setting it to infinity we obtain the Dirac delta, centered at $x_0$. Generally, the Beta distribution has a two-dimensional sufficient statistic $\mathcal{T}(x) = \begin{pmatrix} \log x \\ \log(1 - x) \end{pmatrix}$. However, under this parameterization, we can derive a one-dimensional tail statistic:

$$q(x_t|x_0) \propto \exp\left\{(\alpha_t - 1)\log x_t + (\beta_t - 1)\log(1 - x_t)\right\} = \exp\left\{\nu_t x_0 \log x_t + \nu_t(1 - x_0)\log(1 - x_t)\right\} = \tag{107}$$

$$= \exp\left\{\nu_t x_0 \log \frac{x_t}{1 - x_t} + \nu_t \log(1 - x_t)\right\} = \exp\left\{\eta_t(x_0)\mathcal{T}(x_t) + \log h_t(x_t)\right\} \tag{108}$$

Therefore we can use $\eta_t(x_0) = \nu_t x_0$ and $\mathcal{T}(x) = \log \frac{x}{1-x}$ to define the tail statistic:

$$\mathcal{G}_t(x_{t:T}) = \sum_{s=t}^{T} \nu_s \log \frac{x_s}{1 - x_s} \tag{109}$$

The KL divergence can then be calculated as follows:

$$D_{KL}\left(q^{\text{SS}}(x_t|x_0) \,\|\, p_\theta^{\text{SS}}(x_t|G_t)\right) = \log \frac{\text{Beta}(\alpha_t(x_\theta), \beta_t(x_\theta))}{\text{Beta}(\alpha_t(x_0), \beta_t(x_0))} + \nu_t(x_0 - x_\theta)(\psi(\alpha_t(x_0)) - \psi(\beta_t(x_0))) \tag{110}$$

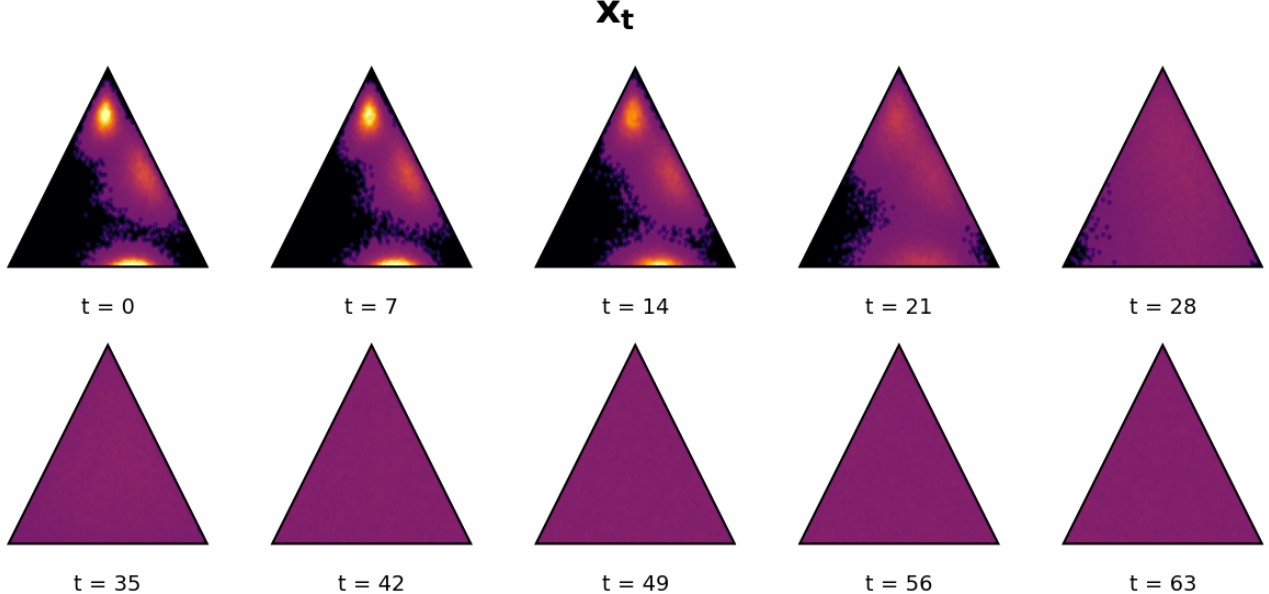

Figure 8: Visualization of the forward process in Dirichlet SS-DDPM on a three-dimensional probabilistic simplex.

### F.3  Dirichlet

$$q(x_t|x_0) = \text{Dirichlet}(x_t; \alpha_t^1, \ldots, \alpha_t^K) \tag{111}$$

Similarly to the Beta distribution, we choose to fix the mode of the distribution at $x_0$ and introduce a concentration parameter $\nu_t$, parameterizing $\alpha_t^k = 1 + \nu_t x_0^k$. This corresponds to using the natural parameter $\eta_t(x_0) = \nu_t x_0$. By setting $\nu_t$ to zero, we recover a uniform distribution, and by setting it to infinity we obtain the Dirac delta, centered at $x_0$. We can use the sufficient statistic $\mathcal{T}(x) = (\log x^1, \ldots, \log x^K)^\mathsf{T}$ to define the tail statistic:

$$\mathcal{G}_t(x_{t:T}) = \sum_{s=t}^{T} \nu_s (\log x_s^1, \ldots, \log x_s^K)^\mathsf{T} \tag{112}$$

The KL divergence can then be calculated as follows:

$$D_{KL}\left(q^{\text{ss}}(x_t|x_0) \,\|\, p_\theta^{\text{ss}}(x_t|G_t)\right) = \sum_{k=1}^{K} \left[ \log \frac{\Gamma(\alpha_t^k(x_\theta))}{\Gamma(\alpha_t^k(x_0))} + \nu_t(x_0^k - x_\theta^k)\psi(\alpha_t^k(x_0)) \right] \tag{113}$$

### F.4  Categorical

$$q(x_t|x_0) = \text{Cat}(x_t; p_t) \tag{114}$$

In this case, we mimic the definition of the categorical diffusion model, used in D3PM. The noising process is parameterized by the probability vector $p_t = x_0 \overline{Q}_t$. By setting $\overline{Q}_0$ to identity, we recover the Dirac delta, centered at $x_0$.

In this parameterization, the natural parameter admits linearization (after some notation abuse):

$$\eta_t(x_0) = \log(x_0 \overline{Q}_t) = x_0 \log \overline{Q}_t, \text{ where} \tag{115}$$

log is taken element-wise, and we assume $0 \log 0 = 0$.

The sufficient statistic here is a vector $x_t^\mathsf{T}$. Therefore, the tail statistic can be defined as follows:

$$\mathcal{G}_t(x_{t:T}) = \sum_{s=t}^{T} \left( \log \overline{Q}_s \cdot x_s^\mathsf{T} \right) = \sum_{s=t}^{T} \log(\overline{Q}_s x_s^\mathsf{T}) \tag{116}$$

The KL divergence can then be calculated as follows:

$$D_{KL}\left(q^{\text{ss}}(x_t|x_0) \,\|\, p_\theta^{\text{ss}}(x_t|G_t)\right) = \sum_{i=1}^{D} (x_0 \overline{Q}_t)_i \log \frac{(x_0 \overline{Q}_t)_i}{(x_\theta \overline{Q}_t)_i} \tag{117}$$

Note that, unlike in the Gaussian case, the categorical star-shaped diffusion is not equivalent to the categorical DDPM. The main difference here is that the input $G_t$ to the predictive model $x_\theta(G_t)$ is now a continuous vector instead of a one-hot vector.

As discussed in Section H, proper normalization is important for training SS-DDPMs. In the case of Categorical distributions, we can use the $\mathrm{SoftMax}(\cdot)$ function to normalize the tail statistics without breaking sufficiency:

$$\tilde{G}_t = \mathrm{SoftMax}\left(\mathcal{G}_t(x_{t:T})\right) \tag{118}$$

To see this, we can retrace the steps of the proof of Theorem 1:

$$q^{\mathrm{ss}}(x_0|x_{t:T}) \propto q(x_0)\exp\{x_0 G_t\} = q(x_0)\exp\left\{x_0 \log \tilde{G}_t + \log \sum_{i=1}^d \exp\left(G_t\right)_i\right\} \propto q(x_0)\exp\left\{x_0 \log \tilde{G}_t\right\} \tag{119}$$

Therefore,

$$q^{\mathrm{ss}}(x_0|x_{t:T}) = \frac{q(x_0)\exp\left\{x_0 \log \tilde{G}_t\right\}}{\sum_{\tilde{x}_0} q(\tilde{x}_0)\exp\left\{\tilde{x}_0 \log \tilde{G}_t\right\}} = q^{\mathrm{ss}}(x_0|\tilde{G}_t) \tag{120}$$

Also, Categorical distributions admit a convenient way to compute fractions involving $q(G_t|x_0)$:

$$q(G_t|x_0) = \sum_{x_{t:T}}\left(\prod_{s\geq t} q(x_s|x_0)\right)\mathbb{1}\left[G_t = \mathcal{G}_t(x_{t:T})\right] = \tag{121}$$

$$= \sum_{x_{t:T}}\exp\left\{\sum_{s\geq t} x_0 \log(\overline{Q}_s)x_s^\mathsf{T}\right\}\mathbb{1}\left[G_t = \mathcal{G}_t(x_{t:T})\right] = \sum_{x_{t:T}}\exp\{x_0 G_t\}\mathbb{1}\left[G_t = \mathcal{G}_t(x_{t:T})\right] = \tag{122}$$

$$= \exp\{x_0 G_t\}\sum_{x_{t:T}}\mathbb{1}\left[G_t = \mathcal{G}_t(x_{t:T})\right] = \exp\{x_0 G_t\}\cdot \#_{G_t} \tag{123}$$

This allows us to define the likelihood term $p(x_0|x_{1:T})$ as follows:

$$p(x_0|G_1) = \frac{q(G_1|x_0)\tilde{p}(x_0)}{\sum_{\tilde{x}_0} q(G_1|\tilde{x}_0)\tilde{p}(\tilde{x}_0)} = \frac{\exp\{x_0 G_1\}\tilde{p}(x_0)}{\sum_{\tilde{x}_0}\exp\{\tilde{x}_0 G_1\}\tilde{p}(\tilde{x}_0)}, \tag{124}$$

where $\tilde{p}(x_0)$ can be defined as the frequency of the token $x_0$ in the dataset.

It also allows us to estimate the mutual information between $x_0$ and $G_t$:

$$I(x_0; G_t) = \mathbb{E}_{q(x_0)q(G_t|x_0)}\log \frac{q(G_t|x_0)}{q(G_t)} = \mathbb{E}_{q(x_0)q(G_t|x_0)}\log \frac{q(G_t|x_0)}{\sum_{\tilde{x}_0} q(G_t|\tilde{x}_0)q(\tilde{x}_0)} = \tag{125}$$

$$= \mathbb{E}_{q(x_0)q(G_t|x_0)}\log \frac{\exp\{x_0 G_t\}\cdot \#_{G_t}}{\sum_{\tilde{x}_0}\exp\{\tilde{x}_0 G_t\}\cdot \#_{G_t}\cdot q(\tilde{x}_0)} = \mathbb{E}_{q(x_0)q(G_t|x_0)}\log \frac{\exp\{x_0 G_t\}}{\sum_{\tilde{x}_0}\exp\{\tilde{x}_0 G_t\}q(\tilde{x}_0)} \tag{126}$$

It can be estimated using Monte Carlo. We use it when defining the noising schedule for Categorical SS-DDPM.

### F.5 von Mises

$$q(x_t|x_0) = \mathrm{vonMises}(x_t; x_0, \kappa_t) \tag{127}$$

The von Mises distribution has two parameters, the mode $x$ and the concentration $\kappa$. It is natural to set the mode of the noising distribution to $x_0$ and vary the concentration parameter $\kappa_t$. When $\kappa_t$ goes to infinity, the von Mises distribution approaches the Dirac delta, centered at $x_0$. When $\kappa_t$ goes to 0, it approaches a uniform distribution on a unit circle. The sufficient statistic is $\mathcal{T}(x_t) = \begin{pmatrix}\cos x_t \\ \sin x_t\end{pmatrix}$, and the corresponding natural parameter is $\eta_t(x_0) = \kappa_t \begin{pmatrix}\cos x_0 \\ \sin x_0\end{pmatrix}$. The tail statistic $\mathcal{G}_t(x_{t:T})$ is therefore defined as follows:

$$\mathcal{G}_t(x_{t:T}) = \sum_{s=t}^T \kappa_s \begin{pmatrix}\cos x_s \\ \sin x_s\end{pmatrix} \tag{128}$$

The KL divergence term can be calculated as follows:

$$D_{KL}\left(q^{\mathrm{ss}}(x_t|x_0)\,\|\,p_\theta^{\mathrm{ss}}(x_t|G_t)\right) = \kappa_t \frac{I_1(\kappa_t)}{I_0(\kappa_t)}(1 - \cos(x_0 - x_\theta)) \tag{129}$$

$$\mathbf{x_t}$$

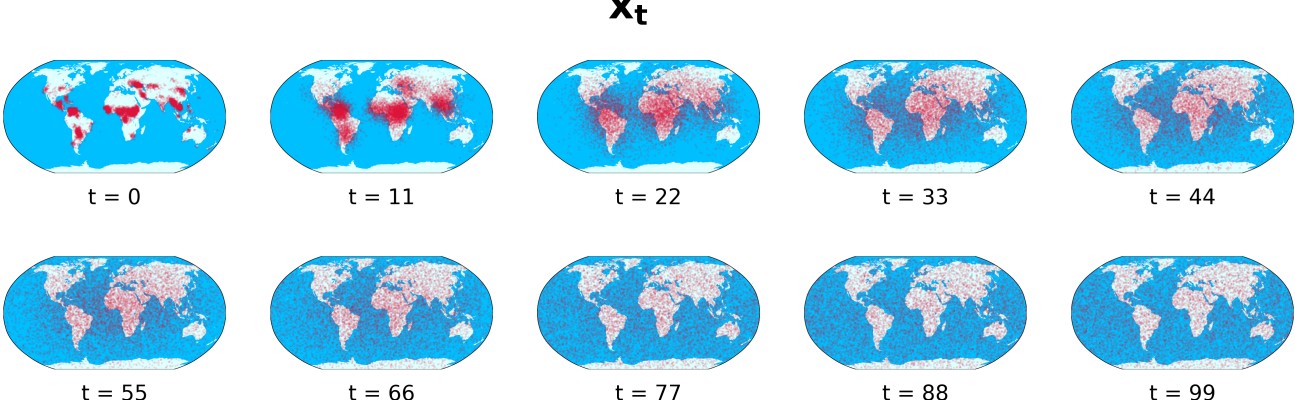

Figure 9: Visualization of the forward process of the von Mises–Fisher SS-DDPM on the three-dimensional unit sphere.

## F.6 von Mises–Fisher

$$q(x_t|x_0) = \text{vMF}(x_t; x_0, \kappa_t) \tag{130}$$

Similar to the one-dimensional case, we set the mode of the distribution to $x_0$, and define the schedule using the concentration parameter $\kappa_t$. When $\kappa_t$ goes to infinity, the von Mises–Fisher distribution approaches the Dirac delta, centered at $x_0$. When $\kappa_t$ goes to 0, it approaches a uniform distribution on a unit sphere. The sufficient statistic is $\mathcal{T}(x) = x$, and the corresponding natural parameter is $\eta_t(x_0) = \kappa_t x_0$. The tail statistic $\mathcal{G}_t(x_{t:T})$ is therefore defined as follows:

$$\mathcal{G}_t(x_{t:T}) = \sum_{s=t}^{T} \kappa_s x_s \tag{131}$$

The KL divergence term can be calculated as follows:

$$D_{KL}\left(q^{\text{ss}}(x_t|x_0) \,\|\, p_\theta^{\text{ss}}(x_t|G_t)\right) = \kappa_t \frac{I_{K/2}(\kappa_t)}{I_{K/2-1}(\kappa_t)} x_0^\mathsf{T}(x_0 - x_\theta) \tag{132}$$

## F.7 Gamma

$$q(x_t|x_0) = \Gamma(x_t; \alpha_t, \beta_t) \tag{133}$$

There are many ways to define a schedule. We choose to interpolate the mean of the distribution from $x_0$ at $t = 0$ to 1 at $t = T$. This can be achieved with the following parameterization:

$$\beta_t(x_0) = \alpha_t(\xi_t + (1 - \xi_t)x_0^{-1}) \tag{134}$$

The mean of the distribution is $\frac{\alpha_t}{\beta_t}$, and the variance is $\frac{\alpha_t}{\beta_t^2}$. Therefore, we recover the Dirac delta, centered at $x_0$, when we set $\xi_t$ to 0 and $\alpha_t$ to infinity. To achieve some standard distribution that doesn't depend on $x_0$, we can set $\xi_t$ to 1 and $\alpha_t$ to some fixed value $\alpha_T$.

In this parameterization, the natural parameters are $\alpha_t - 1$ and $-\beta_t(x_0)$, and the corresponding sufficient statistics are $\log x_t$ and $x_t$. Since the parameter $\alpha_t$ doesn't depend on $x_0$, we only need the sufficient statistic $\mathcal{T}(x_t) = x_t$ to define the tail statistic:

$$\mathcal{G}(x_{t:T}) = \sum_{s=t}^{T} \alpha_s(1 - \xi_s)x_s \tag{135}$$

The KL divergence can be computed as follows:

$$D_{KL}\left(q^{\text{ss}}(x_t|x_0) \,\|\, p_\theta^{\text{ss}}(x_t|G_t)\right) = \alpha_t \left[\log \frac{\beta_t(x_0)}{\beta_t(x_\theta)} + \frac{\beta_t(x_\theta)}{\beta_t(x_0)} - 1\right] \tag{136}$$

**x_t**

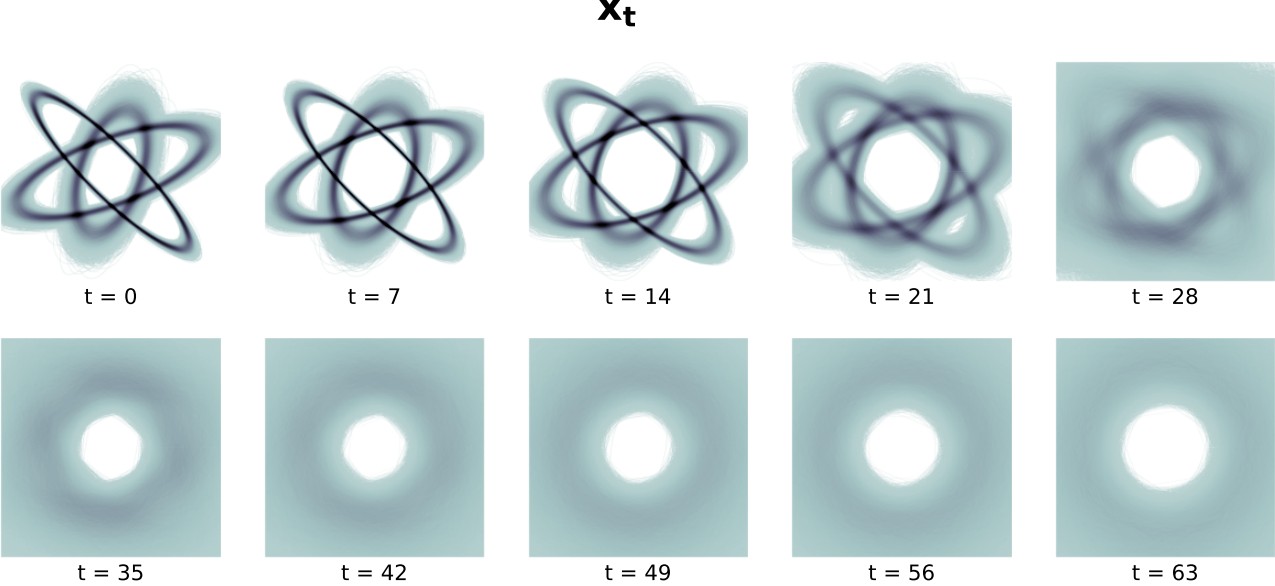

Figure 10: Visualization of the forward process in Wishart SS-DDPM on positive definite matrices of size $2 \times 2$.

## F.8 Wishart

$$q(X_t|X_0) = \mathcal{W}_p(X_t; V_t, n_t) \tag{137}$$

The natural parameters for the Wishart distribution are $-\frac{1}{2}V_t^{-1}$ and $\frac{n_t - p - 1}{2}$. To achieve linear parameterization, we need to linearly parameterize the inverse of $V_t$ rather than $V_t$ directly. Similar to the Gamma distribution, we interpolate the mean of the distribution from $X_0$ at $t = 0$ to $I$ at $t = T$. This can be achieved with the following parameterization:

$$\mu_t(X_0) = \xi_t I + (1 - \xi_t)X_0^{-1} \tag{138}$$
$$V_t(X_0) = n_t^{-1}\mu_t^{-1}(X_0) \tag{139}$$

We recover the Dirac delta, centered at $X_0$, when we set $\xi_t$ to 0 and $n_t$ to infinity. To achieve some standard distribution that doesn't depend on $X_0$, we set $\xi_t$ to 1 and $n_t$ to some fixed value $n_T$.

The sufficient statistic for this distribution is $\mathcal{T}(X_t) = \begin{pmatrix} X_t \\ \log |X_t| \end{pmatrix}$. Since the parameter $n_t$ doesn't depend on $X_0$, we don't need the corresponding sufficient statistic $\log |X_t|$ and can use $\mathcal{T}(X_t) = X_t$ to define the tail statistic:

$$\mathcal{G}(X_{t:T}, t) = \sum_{s=t}^{T} n_s(1 - \xi_s)X_s \tag{140}$$

The KL divergence can then be calculated as follows:

$$D_{KL}\left(q^{\mathrm{ss}}(x_t|x_0) \,\|\, p_\theta^{\mathrm{ss}}(x_t|G_t)\right) = -\frac{n_t}{2}\left[\log\left|V_t^{-1}(X_\theta)V_t(X_0)\right| - \mathrm{tr}\left(V_t^{-1}(X_\theta)V_t(X_0)\right) + p\right] \tag{141}$$

## G  Choosing the noise schedule

In DDPMs, we train a neural network to predict $x_0$ given the current noisy sample $x_t$. In SS-DDPMs, we use the tail statistic $G_t$ as the neural network input instead. Therefore, it is natural to search for a schedule, where the "level of noise" in $G_t$ is similar to the "level of noise" in $x_t$ from some DDPM model. Similarly to D3PM, we formalize the "level of noise" as the mutual information between the clean and noisy samples. For SS-DDPM it would be $I^{\mathrm{ss}}(x_0; G_t)$, and for DDPM it would be $I^{\mathrm{DDPM}}(x_0; x_t)$. We would like to start from a well-performing DDPM model and define a similar schedule by matching $I^{\mathrm{ss}}(x_0; G_t) \approx I^{\mathrm{DDPM}}(x_0; x_t)$. Since Gaussian SS-DDPM is equivalent to DDPM, the desired schedule can be found in Theorem 2. In other cases, however, it is difficult to find a matching schedule.

In our experiments, we found the following heuristic to work well enough. First, we find a Gaussian SS-DDPM schedule that is equivalent to the DDPM with the desired schedule. We denote the corresponding mutual information as $I_{\mathcal{N}}^{\text{ss}}(x_0; G_t) = I^{\text{DDPM}}(x_0; x_t)$. Then, we can match it in the original space of $x_t$ and hope that the resulting mutual information in the space of tail statistics is close too:

$$I^{\text{ss}}(x_0; x_t) \approx I_{\mathcal{N}}^{\text{ss}}(x_0; x_t) \overset{?}{\Rightarrow} I^{\text{ss}}(x_0; G_t) \approx I_{\mathcal{N}}^{\text{ss}}(x_0; G_t) \tag{142}$$

Assuming the schedule is parameterized by a single parameter $\nu$, we can build a look-up table $I^{\text{ss}}(x_0; x_\nu)$ for a range of parameters $\nu$. Then we can use binary search to build a schedule to match the mutual information $I^{\text{ss}}(x_0; x_t)$ to the mutual information schedule $I_{\mathcal{N}}^{\text{ss}}(x_0; x_t)$. While this procedure doesn't allow to match the target schedule *exactly*, it provides a good enough approximation and allows to obtain an adequate schedule. We used this procedure to find the schedule for the Beta diffusion in our experiments with image generation.

To build the look-up table, we need a robust way to estimate the mutual information. The target mutual information $I_{\mathcal{N}}^{\text{ss}}(x_0; x_t)$ can be computed analytically when the data $q(x_0)$ follows a Gaussian distribution. When the data follows an arbitrary distribution, it can be approximated with a Gaussian mixture and the mutual information can be calculated using numerical integration. Estimating the mutual information for arbitrary noising distributions is more difficult. We find that the Kraskov estimator (Kraskov et al., 2004) works well when the mutual information is high ($I > 2$). When the mutual information is lower, we build a different estimator using DSIVI bounds (Molchanov et al., 2019).

$$I^{\text{ss}}(x_0; x_\nu) = D_{KL}\left(q^{\text{ss}}(x_0, x_t) \,\|\, q^{\text{ss}}(x_0) q^{\text{ss}}(x_t)\right) = \mathcal{H}[x_t] - \mathcal{H}[x_t|x_0] \tag{143}$$

This conditional entropy is available in closed form for many distributions in the exponential family. Since the marginal distribution $q^{\text{ss}}(x_0, x_t) = \int q^{\text{ss}}(x_t|x_0)q(x_0)dx_0$ is a semi-implicit distribution (Yin & Zhou, 2018), we can use the DSIVI sandwich (Molchanov et al., 2019) to obtain an upper and lower bound on the marginal entropy $\mathcal{H}[x_t]$:

$$\mathcal{H}[x_t] = -\mathbb{E}_{q^{\text{ss}}} \log q^{\text{ss}}(x_t) = -\mathbb{E}_{q^{\text{ss}}} \log \int q^{\text{ss}}(x_t|x_0)q(x_0)dx_0 \tag{144}$$

$$\mathcal{H}[x_t] \geq -\mathbb{E}_{x_0^{0:K} \sim q(x_0)} \mathbb{E}_{x_t \sim q(x_t|x_0)|_{x_0 = x_0^0}} \log \frac{1}{K+1} \sum_{k=0}^{K} q^{\text{ss}}(x_t|x_0^k) \tag{145}$$

$$\mathcal{H}[x_t] \leq -\mathbb{E}_{x_0^{0:K} \sim q(x_0)} \mathbb{E}_{x_t \sim q(x_t|x_0)|_{x_0 = x_0^0}} \log \frac{1}{K} \sum_{k=1}^{K} q^{\text{ss}}(x_t|x_0^k) \tag{146}$$

These bounds are asymptotically exact and can be estimated using Monte Carlo. We use $K = 1000$ when the mutual information is high ($\frac{1}{2} \leq I < 2$), $K = 100$ when the mutual information is lower ($0.002 \leq I < \frac{1}{2}$), and estimate the expectations using $M = 10^8 K^{-1}$ samples for each timestamp. For values $I > 2$ we use the Kraskov estimator with $M = 10^5$ samples and $k = 10$ neighbors. For values $I < 0.002$ we fit an exponential curve $i(t) = e^{at+b}$ to interpolate between the noisy SIVI estimates, obtained with $K = 50$ and $M = 10^5$.

For evaluating the mutual information $I(x_0; G_t)$ between the clean data and the tail statistics, we use the Kraskov estimator with $k = 10$ and $M = 10^5$.

The mutual information look-up table for the Beta star-shaped diffusion, as well as the used estimations of the mutual information, are presented in Figure 11. The resulting schedule for the beta diffusion is presented in Figure 12, and the comparison of the mutual information schedules for the tail statistics between Beta SS-DDPM and the referenced Gaussian SS-DDPM is presented in Figure 13.

For Categorical SS-DDPM we estimate the mutual information using Monte Carlo. We then choose the noising schedule to match the cosine schedule used by Hoogeboom et al. (2021) using a similar technique.

## H Normalizing the tail statistics

We illustrate different strategies for normalizing the tail statistics in Figure 14. Normalizing by the sum of coefficients is not enough, therefore we resort to matching the mean and the variance empirically. We refer to this trick as time-dependent tail normalization.

Also, proper normalization allows us to visualize the tail statistics by projecting them back into the original domain using $\mathcal{T}^{-1}(\tilde{G}_t)$. The effect of normalization is illustrated in Figure 15.

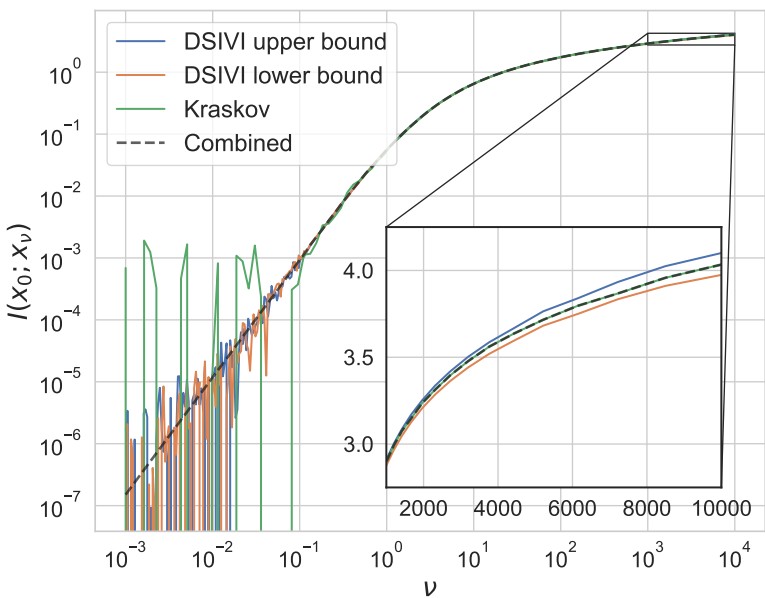

Figure 11: The mutual information look-up table for Beta star-shaped diffusion.

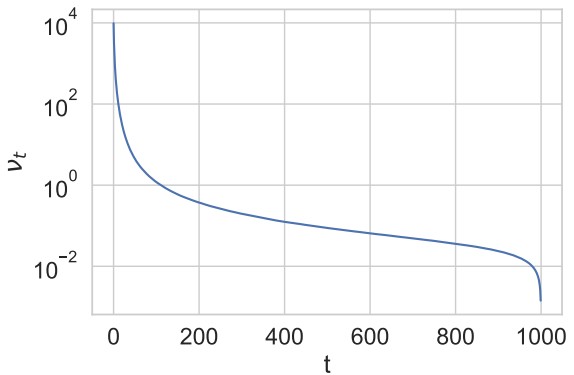

Figure 12: The schedule for Beta star-shaped diffusion.

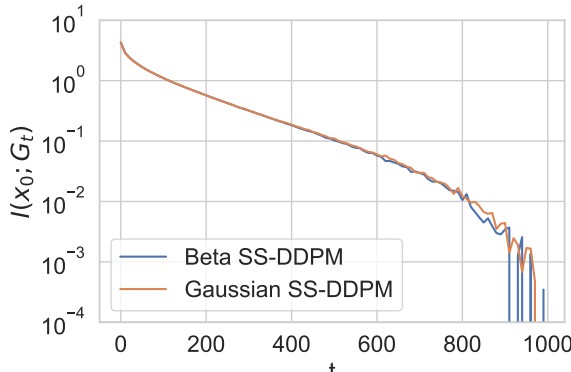

Figure 13: Mutual information between clean data and the tail statistics for beta star-shaped diffusion.

## I  Synthetic data

We compare the performance of DDPM and SS-DDPM on synthetic tasks with exotic data domains. In these tasks, we train and sample from DDPM and SS-DDPM using $T = 64$ steps. We use an MLP with 3 hidden layers of size 512, swish activations and residual connections through hidden layers (He et al., 2015). We use sinusoidal positional time embeddings (Vaswani et al., 2017) of size 32 and concatenate them with the normalized tail statistics $\tilde{G}_t$. We add a mapping to the corresponding domain on top of the network. During training, we use gradient clipping and EMA weights to improve stability. All models on synthetic data were trained for 350k iterations with batch size 128. In all our experiments with SS-DDPM on synthetic data we use time-dependent tail normalization. We use DDPM with linear schedule and $L_{simple}$ or $L_{vlb}$ as a loss function. We choose between $L_{simple}$ and $L_{vlb}$ objective based on the KL divergence between the data distribution and the model distribution $D_{KL}\left(q(x_0) \,\|\, p_\theta(x_0)\right)$. To make an honest comparison, we precompute normalization statistics for DDPM in the same way that we do in time-dependent tail normalization. In DDPM, a neural network makes predictions using $x_t$ as an input, so we precompute normalization statistics for $x_t$ and fix them during the training and sampling stages.

**Probabilistic simplex**  We evaluate Dirichlet SS-DDPM on a synthetic problem of generating objects on a three-dimensional probabilistic simplex. We use a mixture of three Dirichlet distributions with different parameters as training

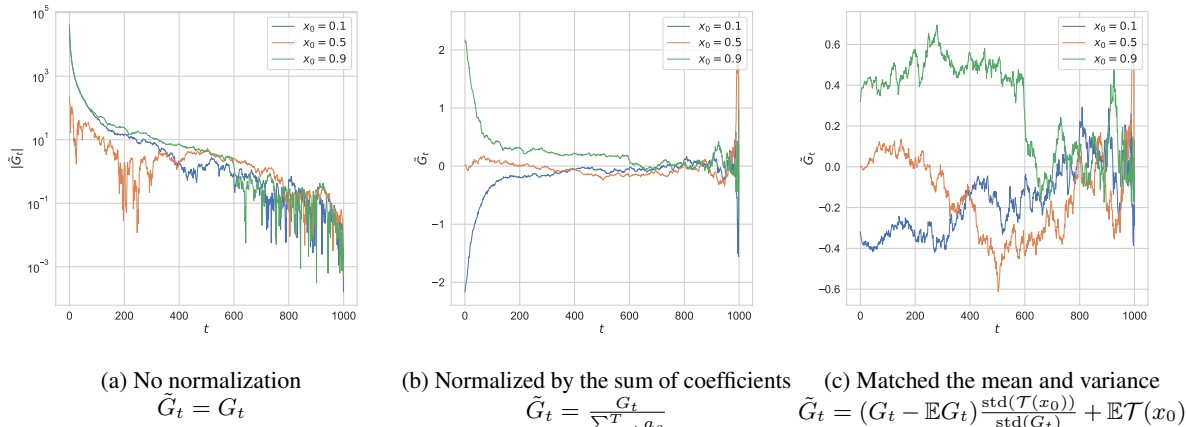

(a) No normalization
$$\tilde{G}_t = G_t$$

(b) Normalized by the sum of coefficients
$$\tilde{G}_t = \frac{G_t}{\sum_{s=t}^{T} a_s}$$

(c) Matched the mean and variance
$$\tilde{G}_t = (G_t - \mathbb{E}G_t)\frac{\text{std}(\mathcal{T}(x_0))}{\text{std}(G_t)} + \mathbb{E}\mathcal{T}(x_0)$$

Figure 14: Example trajectories of the normalized tail statistics with different normalization strategies. The trajectories are coming from a single dimension of Beta SS-DDPM.

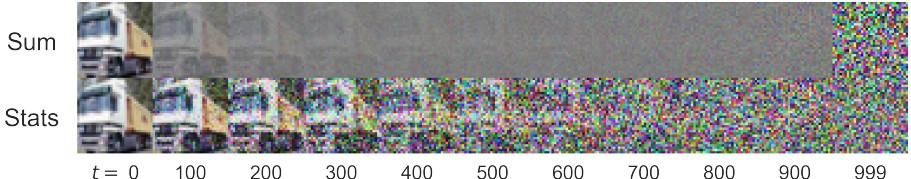

Figure 15: Visualizing the tail statistics for Beta SS-DDPM with different normalization by mapping the normalized tail statistics $\tilde{G}_t$ into the data domain using $\mathcal{T}^{-1}(\tilde{G}_t)$. Top row: normalized by the sum of coefficients, $\tilde{G}_t = \frac{G_t}{\sum_{s=t}^{T} a_s}$. Bottom row: matched the mean and variance, $\tilde{G}_t = (G_t - \mathbb{E}G_t)\frac{\text{std}(\mathcal{T}(x_0))}{\text{std}(G_t)} + \mathbb{E}\mathcal{T}(x_0)$

data. The Dirichlet SS-DDPM forward process is illustrated in Figure 8. To map the predictions to the domain, we put the Softmax function on the top of the MLP. We optimize Dirichlet SS-DDPM on the VLB objective without any modifications using Adam with a learning rate of 0.0004. The DDPM was trained on $L_{vlb}$ using Adam with a learning rate of 0.0002. An illustration of the samples is presented in Table 4.

Table 4: Generating objects from three-dimensional probabilistic simplex.

| Original | DDPM | Dirichlet SS-DDPM |
|---|---|---|

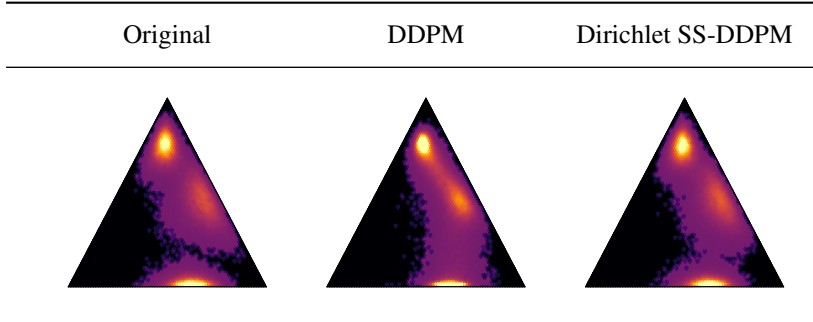

**Symmetric positive definite matrices**  We evaluate Wishart SS-DDPM on a synthetic problem of generating symmetric positive definite matrices of size $2 \times 2$. We use a mixture of three Wishart distributions with different parameters as training data. The Wishart SS-DDPM forward processes are illustrated in Figure 10. For the case of symmetric positive definite matrices $V$, MLP predicts a lower triangular factor $L_\theta$ from the Cholesky decomposition $V_\theta = L_\theta L_\theta^\top$. For stability of sampling from the Wishart distribution and estimation of the loss function, we add a scalar matrix $10^{-4}\mathbf{I}$ to the predicted symmetric positive definite matrix $V_\theta$. In Wishart SS-DDPM we use an analogue of $L_{simple}$ as a loss function. To improve the training stability, we divide each KL term corresponding to a timestamp $t$ by $n_t$, the de-facto concentration parameter

from the used noising schedule (see Table 6). Wishart SS-DDPM was trained using Adam with a learning rate of 0.0004. The DDPM was trained on $L_{simple}$ using Adam with a learning rate of 0.0004. Since positive definite $2 \times 2$ matrices can be interpreted as ellipses, we visualize the samples by drawing the corresponding ellipses. An illustration of the samples is presented in Table 5.

Table 5: Generating symmetric positive definite $2 \times 2$ matrices.

| Original | DDPM | Wishart SS-DDPM |
|----------|------|-----------------|

## J   Geodesic data

We evaluate von Mises–Fisher SS-DDPM on the problem of restoring the distribution on the sphere from empirical data of fires on the Earth's surface (EOSDIS, 2020). We illustrate points on the sphere using a 2D projection of the Earth's map. We train and sample from von Mises–Fisher SS-DDPM using $T = 100$ steps. The forward process is illustrated in Figure 9. We use the same MLP architecture as described in Appendix I and also use time-dependent tail normalization. To map the prediction onto the unit sphere, we normalize the three-dimensional output of the MLP. We optimize $L_{vlb}$ using the AdamW optimizer with a learning rate of 0.0002 and exponential decay with $\gamma = 0.999997$. The model is trained for $2,000,000$ iterations with batch size 100. For inference, we also use EMA weights with a decay of 0.9999.

## K   Discrete data

We evaluate Categorical SS-DDPM on the task of unconditional character-level text generation on the `text8` dataset. We evaluate our model on sequences of length 256. Using the property of Categorical SS-DDPM, that we can directly compute mutual information between $x_0$ and $G_t$ (see Appendix F.4), we match the noising schedule of $G_t$ to the noising schedule of $x_t$ by Austin et al. (2021). We optimize Categorical SS-DDPM on the VLB objective. Following Austin et al. (2021), we use the default T5 encoder architecture with 12 layers, 12 heads, mlp dim 3072 and qkv dim 768. We add positional embeddings to the sequence of tail statistics $G_t$ and also add the sinusoidal positional time embedding to the beginning. We use Adam (Kingma & Ba, 2015) with learning rate $5 \times 10^{-4}$ with a $10,000$-step learning rate warmup, but instead of inverse sqrt decay, we use exponential decay with $\gamma = 0.999995$. We use a standard $90,000,000/5,000,000/500,000$ train-test-validation split and train neural network for 512 epochs (time costs when using 3 NVIDIA A100 GPUs: training took approx. 112 hours and estimating NLL on the test set took approx. 2.5 hours). Since the `Softmax` function does not break the sufficiency of tail statistic in Categorical SS-DDPMs (see Appendix F.4), we can use the `Softmax` function as an efficient input normalization for the neural network. Categorical SS-DDPM also provides us with a convenient way to define $\log p(x_0|G_1)$, and we make use of it to estimate the NLL score (see Appendix F.4).

## L   Image data

In our experiments with Beta SS-DDPM, we use the NCSN++ neural network architecture and train strategy by Song et al. (2020b) (time costs when using 4 NVIDIA 1080 GPUs: training took approx. 96 hours, sampling of $50,000$ images took approx. 10 hours). We put a sigmoid function on the top of NCSN++ to map the predictions to the data domain. We use time-dependent tail normalization and train Beta SS-DDPM with $T = 1000$. We matched the noise schedule in Beta SS-DDPM to the cosine noise schedule in Gaussian DDPM (Nichol & Dhariwal, 2021) in terms of mutual information, as described in Appendix G.

# M Changing discretization

When sampling from DDPMs, we can skip some timestamps to trade off computations for the quality of the generated samples. For example, we can generate $x_{t_1}$ from $x_{t_2}$ in one step, without generating the intermediate variables $x_{t_1+1:t_2-1}$:

$$\tilde{p}_\theta^{\mathrm{DDPM}}(x_{t_1}|x_{t_2}) = q^{\mathrm{DDPM}}(x_{t_1}|x_{t_2}, x_0)|_{x_0 = x_\theta^{\mathrm{DDPM}}(x_{t_2}, t_2)} = \tag{147}$$

$$= \mathcal{N}\left(x_{t_1}; \frac{\sqrt{\overline{\alpha}_{t_1}^{\mathrm{DDPM}}}\left(1 - \frac{\overline{\alpha}_{t_2}^{\mathrm{DDPM}}}{\overline{\alpha}_{t_1}^{\mathrm{DDPM}}}\right)}{1 - \overline{\alpha}_{t_2}^{\mathrm{DDPM}}} x_\theta^{\mathrm{DDPM}}(x_{t_2}, t_2) + \frac{\sqrt{\frac{\overline{\alpha}_{t_2}^{\mathrm{DDPM}}}{\overline{\alpha}_{t_1}^{\mathrm{DDPM}}}}\left(1 - \overline{\alpha}_{t_1}^{\mathrm{DDPM}}\right)}{1 - \overline{\alpha}_{t_2}^{\mathrm{DDPM}}} x_{t_2}, \frac{1 - \overline{\alpha}_{t_1}^{\mathrm{DDPM}}}{1 - \overline{\alpha}_{t_2}^{\mathrm{DDPM}}}\left(1 - \frac{\overline{\alpha}_{t_2}^{\mathrm{DDPM}}}{\overline{\alpha}_{t_1}^{\mathrm{DDPM}}}\right)\right) = \tag{148}$$

$$= \mathcal{N}\left(x_{t_1}; \frac{\overline{\alpha}_{t_1}^{\mathrm{DDPM}} - \overline{\alpha}_{t_2}^{\mathrm{DDPM}}}{\sqrt{\overline{\alpha}_{t_1}^{\mathrm{DDPM}}}(1 - \overline{\alpha}_{t_2}^{\mathrm{DDPM}})} x_\theta^{\mathrm{DDPM}}(x_{t_2}, t_2) + \frac{\sqrt{\overline{\alpha}_{t_2}^{\mathrm{DDPM}}}(1 - \overline{\alpha}_{t_1}^{\mathrm{DDPM}})}{\sqrt{\overline{\alpha}_{t_1}^{\mathrm{DDPM}}}(1 - \overline{\alpha}_{t_2}^{\mathrm{DDPM}})} x_{t_2}, \frac{(1 - \overline{\alpha}_{t_1}^{\mathrm{DDPM}})(\overline{\alpha}_{t_1}^{\mathrm{DDPM}} - \overline{\alpha}_{t_2}^{\mathrm{DDPM}})}{(1 - \overline{\alpha}_{t_2}^{\mathrm{DDPM}})\overline{\alpha}_{t_1}^{\mathrm{DDPM}}}\right) \tag{149}$$

In the general case of SS-DDPM, we can't just skip the variables. If we skip the variables, the corresponding tail statistics will become atypical and the generative process will fail. To keep the tail statistics adequate, we can sample all the intermediate variables $x_t$ but do it in a way that doesn't use additional function evaluations:

$$G_{t_1} = \sum_{s=t_1}^{t_2-1} A_s^\mathsf{T} \mathcal{T}(x_s) + G_{t_2}, \text{ where} \tag{150}$$

$$x_s \sim q(x_s|x_0)|_{x_0 = x_\theta^{\mathrm{SS}}(G_{t_2}, t_2)} \tag{151}$$

In case of Gaussian SS-DDPM this trick is equivalent to skipping the variables in DDPM:

$$G_{t_1} = \frac{1 - \overline{\alpha}_{t_1}^{\mathrm{DDPM}}}{\sqrt{\overline{\alpha}_{t_1}^{\mathrm{DDPM}}}} \sum_{s=t_1}^{t_2-1} \frac{\sqrt{\overline{\alpha}_s^{\mathrm{SS}}} x_s}{1 - \overline{\alpha}_s^{\mathrm{SS}}} + \frac{1 - \overline{\alpha}_{t_1}^{\mathrm{DDPM}}}{\sqrt{\overline{\alpha}_{t_1}^{\mathrm{DDPM}}}} \frac{\sqrt{\overline{\alpha}_{t_2}^{\mathrm{DDPM}}}}{1 - \alpha_{t_2}^{\mathrm{DDPM}}} G_{t_2} = \tag{152}$$

$$= \frac{1 - \overline{\alpha}_{t_1}^{\mathrm{DDPM}}}{\sqrt{\overline{\alpha}_{t_1}^{\mathrm{DDPM}}}} \sum_{s=t_1}^{t_2-1} \frac{\overline{\alpha}_s^{\mathrm{SS}}}{1 - \overline{\alpha}_s^{\mathrm{SS}}} x_\theta^{\mathrm{SS}}(G_{t_2}, t_2) + \frac{1 - \overline{\alpha}_{t_1}^{\mathrm{DDPM}}}{\sqrt{\overline{\alpha}_{t_1}^{\mathrm{DDPM}}}} \sum_{s=t_1}^{t_2-1} \sqrt{\frac{\overline{\alpha}_s^{\mathrm{SS}}}{1 - \overline{\alpha}_s^{\mathrm{SS}}}} \epsilon_s + \frac{\sqrt{\overline{\alpha}_{t_2}^{\mathrm{DDPM}}}(1 - \overline{\alpha}_{t_1}^{\mathrm{DDPM}})}{\sqrt{\overline{\alpha}_{t_1}^{\mathrm{DDPM}}}(1 - \overline{\alpha}_{t_2}^{\mathrm{DDPM}})} G_{t_2} \tag{153}$$

$$\mathbb{E}G_{t_1} = \frac{1 - \overline{\alpha}_{t_1}^{\mathrm{DDPM}}}{\sqrt{\overline{\alpha}_{t_1}^{\mathrm{DDPM}}}}\left(\frac{\overline{\alpha}_{t_1}^{\mathrm{DDPM}}}{1 - \overline{\alpha}_{t_1}^{\mathrm{DDPM}}} - \frac{\overline{\alpha}_{t_2}^{\mathrm{DDPM}}}{1 - \overline{\alpha}_{t_2}^{\mathrm{DDPM}}}\right) x_\theta^{\mathrm{SS}}(G_{t_2}, t_2) + \frac{\sqrt{\overline{\alpha}_{t_2}^{\mathrm{DDPM}}}(1 - \overline{\alpha}_{t_1}^{\mathrm{DDPM}})}{\sqrt{\overline{\alpha}_{t_1}^{\mathrm{DDPM}}}(1 - \overline{\alpha}_{t_2}^{\mathrm{DDPM}})} G_{t_2} = \tag{154}$$

$$= \frac{(1 - \overline{\alpha}_{t_1}^{\mathrm{DDPM}})(\overline{\alpha}_{t_1}^{\mathrm{DDPM}} - \overline{\alpha}_{t_2})}{\sqrt{\overline{\alpha}_{t_1}^{\mathrm{DDPM}}}(1 - \overline{\alpha}_{t_1}^{\mathrm{DDPM}})(1 - \overline{\alpha}_{t_2}^{\mathrm{DDPM}})} x_\theta^{\mathrm{SS}}(G_{t_2}, t_2) + \frac{\sqrt{\overline{\alpha}_{t_2}^{\mathrm{DDPM}}}(1 - \overline{\alpha}_{t_1}^{\mathrm{DDPM}})}{\sqrt{\overline{\alpha}_{t_1}^{\mathrm{DDPM}}}(1 - \overline{\alpha}_{t_2}^{\mathrm{DDPM}})} G_{t_2} = \tag{155}$$

$$= \frac{(\overline{\alpha}_{t_1}^{\mathrm{DDPM}} - \overline{\alpha}_{t_2})}{\sqrt{\overline{\alpha}_{t_1}^{\mathrm{DDPM}}}(1 - \overline{\alpha}_{t_2}^{\mathrm{DDPM}})} x_\theta^{\mathrm{SS}}(G_{t_2}, t_2) + \frac{\sqrt{\overline{\alpha}_{t_2}^{\mathrm{DDPM}}}(1 - \overline{\alpha}_{t_1}^{\mathrm{DDPM}})}{\sqrt{\overline{\alpha}_{t_1}^{\mathrm{DDPM}}}(1 - \overline{\alpha}_{t_2}^{\mathrm{DDPM}})} G_{t_2} \tag{156}$$

$$\mathbb{D}G_{t_1} = \frac{(1 - \overline{\alpha}_{t_1}^{\mathrm{DDPM}})^2}{\overline{\alpha}_{t_1}^{\mathrm{DDPM}}} \sum_{s=t_1}^{t_2-1} \frac{\overline{\alpha}_s^{\mathrm{SS}}}{1 - \overline{\alpha}_s^{\mathrm{SS}}} = \frac{(1 - \overline{\alpha}_{t_1}^{\mathrm{DDPM}})^2}{\overline{\alpha}_{t_1}^{\mathrm{DDPM}}} \frac{\overline{\alpha}_{t_1}^{\mathrm{DDPM}} - \overline{\alpha}_{t_2}^{\mathrm{DDPM}}}{(1 - \overline{\alpha}_{t_1}^{\mathrm{DDPM}})(1 - \overline{\alpha}_{t_2}^{\mathrm{DDPM}})} = \tag{157}$$

$$= \frac{(1 - \overline{\alpha}_{t_1}^{\mathrm{DDPM}})(\overline{\alpha}_{t_1}^{\mathrm{DDPM}} - \overline{\alpha}_{t_2}^{\mathrm{DDPM}})}{(1 - \overline{\alpha}_{t_2}^{\mathrm{DDPM}})\overline{\alpha}_{t_1}^{\mathrm{DDPM}}} \tag{158}$$

In general case, this trick can be formalized as the following approximation to the reverse process:

$$p_\theta^{\mathrm{SS}}(x_{t_1:t_2}|G_{t_2}) = \prod_{t=t_1}^{t_2} q^{\mathrm{SS}}(x_t|x_0)|_{x_0 = x_\theta(\mathcal{G}_t(x_{t:T}), t)} \approx \prod_{t=t_1}^{t_2} q^{\mathrm{SS}}(x_t|x_0)|_{x_0 = x_\theta(\mathcal{G}_{t_2}(x_{t_2:T}), t_2)} \tag{159}$$

Table 6: Examples of SS-DDPM in different families. There are many different ways to parameterize the distributions and the corresponding schedules. Further details are discussed in Appendix F.1–F.8.

| DISTRIBUTION $q^{\mathrm{SS}}(x_t\|x_0)$ | NOISING SCHEDULE | TAIL STATISTIC $\mathcal{G}_t(x_{t:T})$ | KL DIVERGENCE $D_{KL}\left(q^{\mathrm{SS}}(x_t\|x_0) \,\|\, p_\theta^{\mathrm{SS}}(x_t\|x_{t+1:T})\right)$ |
|---|---|---|---|
| GAUSSIAN $\mathcal{N}\left(x_t; \sqrt{\overline{\alpha}_t}x_0, 1-\overline{\alpha}_t\right)$ $x_t \in \mathbb{R}$ | $1 \xleftarrow[t\to 0]{} \overline{\alpha}_t \xrightarrow[t\to T]{} 0$ | $\frac{1-\overline{\alpha}'_s}{\sqrt{\overline{\alpha}'_s}} \sum_{s=t}^T \frac{\sqrt{\overline{\alpha}_s}x_s}{1-\overline{\alpha}_s}$, where $\overline{\alpha}'_t = \frac{\sum_{s=t}^T \frac{\overline{\alpha}_s}{1-\overline{\alpha}_s}}{1+\sum_{s=t}^T \frac{\overline{\alpha}_s}{1-\overline{\alpha}_s}}$ | $\frac{\overline{\alpha}_t(x_\theta - x_0)^2}{2(1-\overline{\alpha}_t)}$ |
| BETA $\mathrm{Beta}\left(x_t; \alpha_t(x_0), \beta_t(x_0)\right)$ $x_t \in [0,1]$ | $\alpha_t(x_0) = 1 + \nu_t x_0$ $\beta_t(x_0) = 1 + \nu_t(1-x_0)$ $+\infty \xleftarrow[t\to 0]{} \nu_t \xrightarrow[t\to T]{} 0$ | $\sum_{s=t}^T \nu_s \log \frac{x_s}{1-x_s}$ | $\log \frac{\mathrm{Beta}(\alpha_t(x_\theta), \beta_t(x_\theta))}{\mathrm{Beta}(\alpha_t(x_0), \beta_t(x_0))} +$ $+ \nu_t(x_0 - x_\theta)(\psi(\alpha_t(x_0)) - \psi(\beta_t(x_0)))$ |
| DIRICHLET $\mathrm{Dir}\left(x_t; \alpha_t^1(x_0), \ldots, \alpha_t^K(x_0)\right)$ $x_t \in [0,1]^K$ $\sum_{i=1}^K x_t^k = 1$ | $\alpha_t^k(x_0) = 1 + \nu_t x_0^k$ $+\infty \xleftarrow[t\to 0]{} \nu_t \xrightarrow[t\to T]{} 0$ | $\sum_{s=t}^T \nu_s \log x_s$ | $\sum_{k=1}^K \left[ \log \frac{\Gamma(\alpha_t^k(x_\theta))}{\Gamma(\alpha_t^k(x_0))} + \right.$ $\left. + \nu_t(x_0^k - x_\theta^k)\psi(\alpha_t^k(x_0)) \right]$ |
| CATEGORICAL $\mathrm{Cat}\left(x_t; p_t(x_0)\right)$ $x_t \in \{0,1\}^D$ $\sum_{i=1}^D x_t^i = 1$ | $p_t(x_0) = x_0 \overline{Q}_t$ $I \xleftarrow[t\to 0]{} \overline{Q}_t \xrightarrow[t\to T]{} \overline{Q}_T$ | $\sum_{s=t}^T \log\left(\overline{Q}_s x_s^\mathsf{T}\right)$ | $\sum_{i=1}^D (p_t(x_0))_i \log \frac{(p_t(x_0))_i}{(p_t(x_\theta))_i}$ |
| VON MISES $\mathrm{vM}\left(x_t; x_0, \kappa_t\right)$ $x_t \in [-\pi, \pi]$ | $+\infty \xleftarrow[t\to 0]{} \kappa_t \xrightarrow[t\to T]{} 0$ | $\sum_{s=t}^T \kappa_s \begin{pmatrix} \cos x_s \\ \sin x_s \end{pmatrix}$ | $\kappa_t \frac{I_1(\kappa_t)}{I_0(\kappa_t)}(1 - \cos(x_0 - x_\theta))$ |
| VON MISES–FISHER $\mathrm{vMF}\left(x_t; x_0, \kappa_t\right)$ $x_t \in [-1,1]^K$ $\|x_t\| = 1$ | $+\infty \xleftarrow[t\to 0]{} \kappa_t \xrightarrow[t\to T]{} 0$ | $\sum_{s=t}^T \kappa_s x_s$ | $\kappa_t \frac{I_{K/2}(\kappa_t)}{I_{K/2-1}(\kappa_t)} x_0^\mathsf{T}(x_0 - x_\theta)$ |
| GAMMA $\Gamma\left(x_t; \alpha_t, \beta_t(x_0)\right)$ $x_t \in (0, +\infty)$ | $\beta_t(x_0) = \alpha_t(\xi_t + (1-\xi_t)x_0^{-1})$ $+\infty \xleftarrow[t\to 0]{} \alpha_t \xrightarrow[t\to T]{} \alpha_T$ $0 \xleftarrow[t\to 0]{} \xi_t \xrightarrow[t\to T]{} 1$ | $\sum_{s=t}^T \alpha_s(1-\xi_s)x_s$ | $\alpha_t \left[ \log \frac{\beta_t(x_0)}{\beta_t(x_\theta)} + \frac{\beta_t(x_\theta)}{\beta_t(x_0)} - 1 \right]$ |
| WISHART $\mathcal{W}\left(X_t; n_t, V_t(X_0)\right)$ $X_t \in \mathbb{R}^{p\times p}$ $X_t \succ 0$ | $\mu_t(X_0) = \xi_t I + (1-\xi_t)X_0^{-1}$ $V_t(X_0) = n_t^{-1}\mu_t^{-1}(X_0)$ $+\infty \xleftarrow[t\to 0]{} n_t \xrightarrow[t\to T]{} n_T$ $0 \xleftarrow[t\to 0]{} \xi_t \xrightarrow[t\to T]{} 1$ | $\sum_{s=t}^T n_s(1-\xi_s)X_s$ | $-\frac{n_t}{2}\left[ \log \left\|V_t^{-1}(X_\theta)V_t(X_0)\right\| - \right.$ $\left. -\mathrm{tr}\left(V_t^{-1}(X_\theta)V_t(X_0)\right) + p \right]$ |

