# OpenReview forum: "Star-Shaped Denoising Diffusion Probabilistic Models"
_NeurIPS.cc/2023/Conference — NeurIPS 2023 poster_

### Official Review · Reviewer_7FuV · 2023-06-25

**Soundness:** 4 excellent
**Presentation:** 4 excellent
**Contribution:** 4 excellent
**Rating:** 8
**Confidence:** 4

**Summary:**

This introduces the so-called Star-Shaped DDPM (SS-DDPM). Instead of using Gaussian forward diffusion step in a Markovian manner, the diffusion process is directly conditioned on the data, in order to construct the "true" reverse process (posterior), resulting in the star-shaped diffusion process. This opens the door for using other transition process as demonstrated in the paper.

**Strengths:**

1. This is a solid paper that I have reviewed so far.  All the conclusions are backed by the theory.
2. The paper is well presented and easy to follow the main the idea with full materials in the supplementary
3. The idea presented in the paper is novel and may have wide applications.

**Weaknesses:**

Frankly speaking, I like to see this paper to be accepted.  I cannot form any weakness.  The only point I may raise is it would be much much better for the authors to share their implementation source code.

**Questions:**

When using other transition distributions in the forward process, what is the strategy to actually diffuse the data (into noises)?

**Limitations:**

Not much in this perspective. No comments

---

> ### Author Rebuttal · Authors · 2023-08-09
>
> Thank you for the kind comments!
>
> We will release the source code with the camera ready version of the paper.
>
> Regarding your question:
> In general, we follow the same intuition as when designing the forward process in DDPM. We start with low noise ($x_1$, $x_2$, ... should be reasonably close to $x_0$), and end with high noise ($x_T$, $x_{T-1}$, … should be almost pure noise, almost independent from $x_0$). And then we need to choose the probabilities such that the inputs to the denoising neural network (in DDPM it is $x_t$, and in our case it is the sufficient tail statistic $G_t$) gradually becomes more and more noisy. See Figure 5 for reference. To choose the transition probabilities somewhat automatically, we can refer to the rate at which the mutual information $I(x_0; G_t)$ changes with $t$, see Appendix 7: Choosing the noise schedule for details.

---

> > ### Comment · Reviewer_7FuV · 2023-08-10
> >
> > Thanks

---

### Official Review · Reviewer_P24X · 2023-07-03

**Soundness:** 3 good
**Presentation:** 4 excellent
**Contribution:** 3 good
**Rating:** 6
**Confidence:** 4

**Summary:**

The paper introduces a new probabilistic structure for denoising models that does away with a Markov forward process. The authors derive the reverse process in terms of a sufficient statistic that allows for efficient reverse sampling. The form of the model is derived for a variety of noising distributions and the model is evaluated on a range of different data modalities.

**Strengths:**

This work can potentially help many people in the generative modelling community because there has been a lot of work trying to extend corruption process to different data types e.g. discrete, manifold data and this paper has a general recipe that appears to work for a large chunk of these and more which could be really helpful when encountering non-standard data types. The example of the Wishart distribution on positive definite matrices is very cool.

I can foresee many extensions to this work, looking to bring more distributions into the framework, extending it to continuous time which could bring up a lot of interesting technical problems and broadening the class of possible distributions by introducing approximate sufficient statistics.

The paper has a good suite of experiments on a variety of real world practically relevant data such as text and images and the performance of the model seems good

**Weaknesses:**

I think the paper could benefit from more intuitions and visualizations for the role of G. Initially, the model seems strange as x_t can jump a lot between x_t and x_{t+1} due to going to x_0 and resampling q(x_t | x_0). However, it seems that G instead is the smoothly changing statistic that is more analogous to the state in previous diffusion models. For example, I think figure 9 in the supplementary is quite interesting and could be moved to the main as it shows that a correctly normalized G looks very similar to the state in norm diffusion models. It would be good to have more intuition as to the role of G and how it is more closely linked to the 'x_t' of normal diffusion models.

Further, there is not really a experiment that really motivates the use of these more interesting distributions in a scenario where normal diffusion models really don't work at all. I appreciate that this is somewhat an open question at the moment in the literature more generally and the community has already proven an interest in adapting the corruption process to the data modality of interest so this may not be a big requirement.

Edit after rebuttal: I have read the authors rebuttal and appreciate the answers to my questions about the derivations and the FID scores. I intend to keep my score as it is.

**Questions:**

In the proof of Theorem 1, how do you go from line (23) to line (24), as in, how do you know what q(G_t | x_0) is, for the numerator in (24)?

Figure 6 is confusing to me because the figure makes it seem like the model is better than DDPM at all NFE but in the text the best FID is stated to be 3.17 which is the same as the best FID from DDPM and so shouldn't Figure 6 show the method and DDPM at least overlapping at 10^3 sampling steps?

In DDPM, the Gaussian form of the reverse distribution is justified by considering the limit as the noise in each step is taken to 0 and the number of steps is taken to infinity. Is there any such justification for the Gaussian form that you assume in your model or is this a heuristic?

**Limitations:**

The limitations of the framework e.g. restriction of the distributions and ability to find the sufficient statistics was nicely discussed.

---

> ### Author Rebuttal · Authors · 2023-08-09
>
> Thank you for the comments! We would like to address your questions below.
>
> **Weakness 1:**
> We used Figure 5 to illustrate this point in the main text, however we should probably make this point more clear in the text. The close connection between $G_t$ in SS-DDPM and $x_t$ in DDPM is crucial to understanding the model. We would try to expand Figure 5, give more attention to Figure 9 or rewrite the text to make this more clear.
>
> **Question 1:**
> The first transition ((23) to (24)) is just using the normalization constraint. Since we have started with a distribution on $x_0$, we can drop all multiplicative terms that do not depend on $x_0$ and then renormalize. We have started with a conditional distribution $q(x_0 | x_{t:T})$, and were able to rewrite the density as a function of $x_0$ and $G_t$. This immediately means that the resulting distribution is a conditional distribution $q(x_0 | G_t)$, hence the second transition on line (24). Because of that we do not need to compute these distributions explicitly. We have unintentionally skipped over this step as it has previously seemed to be obviously true. However, we were not able to recall or find an existing lemma that establishes this fact (although surely it must exist somewhere), so we will include a more detailed proof. Thanks for pointing it out! Something along these lines should work:
>
> **Lemma 1.** Assume $x$ and $y$ follow a joint distribution $p(x, y) = p(x)p(y | x)$, where $p(y | x)$ is defined as $f(y, h(x))$. We would like to show that $p(y | x)=p(y | z)|_{z=h(x)}$, where $z=h(x)$.
>
> We can write down the joint distribution $p(x, y, z) = p(x)f(y, h(x))\delta(z - h(x))$. Then $p(y, z)=\int p(x, y, z) dx=$ $\int p(x)f(y, h(x))\delta(z - h(x)) dx =$ $\int p(x)f(y, z)\delta(z - h(x)) dx =$ $f(y, z)\int p(x)\delta(z - h(x)) dx=$ $f(y, z)p(z)$.
> Finally, the conditional probability is $p(y | z)=\frac{p(y, z)}{p(z)}=f(y, z)$ — the same function.
>
> In the case of line (24) $x_0$ is $y$, $x_{t..T}$ is $x$ and $G_t$ is $z$.
>
> **Question 2:**
> Unfortunately, the original DDPM paper did not include the results on shortened generation (FID vs NFE). Therefore, we used the results reported by Nichol&Dhariwal 2021 (Improved DDPM), where the final performance of DDPM was slightly worse. This discrepancy is important, which is why we only conclude that SS-DDPM achieves a comparable performance rather than strictly outperforms DDPM.
>
> **Question 3:**
> In our model we use the same distribution for the reverse process as we do for the forward process (e.g. Beta distribution for Beta SS-DDPM, etc — we define it in eq. 20). This is a natural choice because different distributions from the exponential family have different supports, and choosing other distributions would likely make the involved KL divergences infinite. Also, when these distributions are kept the same, it is usually possible to compute the KL divergence analytically and simplify it (see Table 3 in the end of the Appendix). However, you are right that this is just an approximation, and in general other distributions could make sense too. For example, mixtures could be a good candidate to consider: the “true” reverse distribution in eq. (19) is essentially an infinite mixture of these distributions. The same argument could be made for Gaussian discrete-time DDPMs too.

---

> > ### Comment · Reviewer_P24X · 2023-08-11
> >
> > Thank you for the response. I understand now how to get to line (24), I think just the trick of renormalizing wrt x_0 is what I was missing. For the FID results, my worry was that the DDPM results might have been calculated with 10K samples whereas others with 50K and when you use less samples in FID it biases the result upwards. But it seems in Nichol&Dhariwal 2021, they were using 50K for this plot so I think the results still stand. I will keep my score as it is currently.

---

### Official Review · Reviewer_EaR3 · 2023-07-05

**Soundness:** 1 poor
**Presentation:** 2 fair
**Contribution:** 2 fair
**Rating:** 5
**Confidence:** 5

**Summary:**

This paper proposes a star-shaped diffusion probabilistic model, which is non-Markovian, and more like an autoregressive way of predicting intermediate states xt. Specifically, the authors define a forward process q(xt|x0) that each intermediate state xt is directly related to the initial state x0. The reverse process is defined by p(xt|xt, xt+1, ..., xT). To efficiently solve p(xt|xt, xt+1, ..., xT), the authors propose a compressive representation Gt(xt, xt+1, ..., xT) to extract all the information of x0 contained in history states {xt+1, ..., xT}, so the modified distribution becomes p(xt|Gt). The training and inference go in a similar way as DDPM. Simple experiments show that the proposed method performs better than the baseline methods MTD and DDPM.

**Strengths:**

1. This paper provides a novel view of the definition of diffusion models and is very inspirational. It is worth thinking about whether the Markov process is the best design choice considering the fact that the whole history {xt, xt+1, ..., xT} contains more information than a single state xt.

2. The author proposed a novel diffusion model with basic evaluations, which may inspire the community for further exploration.

**Weaknesses:**

1. The presentation needs to be improved: (1) T, Ωt, and ht in Eq.13 lack interpretation; (2) It is not well proved that Gt covers all the information of x0 in {xt, xt+1, ..., xT}. Although the authors mention that there are explanations in the supplementary material, you need to ensure the reader has a basic understanding by just reading the paper.

2. The advantage and application value of the proposed method is not clear. For example, can you provide a practical case that DDPM can not address while SS-DDPM can?

3. The experiments are too simple and not persuasive. To prove the advantage of SS-DDPM, it is encouraged to compare with state-of-the-art methods on diverse benchmarks,

4. As the authors said in line 128, "In general case the dimensionality of Gt would grow with the size of the tail"

4. There is no evidence that SS-DDPM can apply acceleration methods like DDIM.

**Questions:**

I doubt the reverse process of q(xt|x0) should be p(x0|xt), can you explain why the reverse process is p(xt|xt, xt+1, ..., xT)?

**Limitations:**

The scope of this paper is interesting, but it needs more time to polish the theory, presentation, and experiments. Considering its current form, I think this paper does not meet the acceptance criteria of NeurIPS. However, if there is enough evidence to show the value of this paper, I may also change my score.

---

> ### Author Rebuttal · Authors · 2023-08-09
>
> Thank you for the comments! We would like to address your concerns and questions below.
>
> **Weakness 1 (1):**
> Eq. 13 is a standard definition of the exponential family of distributions with a standard notation. $\mathcal{T}(x_t)$ is the sufficient statistic, $\eta(x_0)$ is the natural parameter, $h(x_t)$ is the base measure and $\Omega(x_0)$ is a normalization constant, sometimes called a (log)partition function. We will improve this section.
>
> **Weakness 1 (2):**
> We formulate this property as Theorem 1 and prove it in Appendix 3. It is a technical proof, and it is common practice to only include the proofs in the Appendix. The intuition behind this theorem is the same as behind the original PKD theorem. We generalize it for the non-i.i.d. case, which is why we have to add an additional constraint (linearity of the parameterization). Strictly speaking, we prove the following equality: $q(x_0 | x_t, \dots, x_T) = q(x_0 | G_t)$, which means that “$G_t$ covers all the information of $x_0$ in $x_t,\dots, x_T$”.
>
> **Weakness 2:**
> When the data or its parts lie on a manifold, its density in the original space would be degenerate, and DDPM would fail to recover it. Even if DDPM would produce good-looking samples, its density estimation would be unreliable. On the contrary, SS-DDPM with an appropriate distribution would both produce good-looking samples and a reliable estimate of the density. For example, data like molecular graphs could combine categorical data (atoms), spherical data (orientations) and positive data (distances).
>
> **Weakness 3:**
> At this stage our goal was to demonstrate that the model can be successfully applied with a variety of noising distributions rather than perfecting the model for each individual task or finding the best task for the model. Comparing with domain-specific state-of-the-art methods would require ad-hoc task-specific modifications and a very expensive hyperparameter search to make the comparison fair. These more challenging settings are something we would like to explore in the future.
>
> **Weakness 4:**
> In this paper we only consider distributions from the exponential family. In that case we provide a simple recipe for constructing a sufficient tail statistic $G_t$ of fixed dimensionality. Although the general case is beyond the scope of our paper, we suspect that in practice adequate statistics of fixed size would be enough, even if they are not strictly sufficient statistics.
>
> **Weakness 5:**
> This paper is focused on establishing the base SS-DDPM model. Since SS-DDPM and DDPM are closely connected, we suspect that it would be possible to adapt other modifications of DDPM to SS-DDPM as well (e.g. DDIM, DDRM, connection to SDE). However, these extensions are beyond the scope of this paper.
>
> **Question 1:**
> We discuss and illustrate it in detail in Appendix 2. The short answer is that we need to look at the forward and reverse processes as a whole, not at inverting individual steps.
>
> Longer answer: Diffusion models model the whole set of variables $x_0, x_1, \dots, x_T$ by learning a joint distribution $p_\theta(x_0, x_1, \dots, x_T)$, called the reverse process, to approximate a fixed joint distribution $q(x_0, x_1, \dots, x_T)$, called the forward process. When the forward process is Markovian, i.e. it factorizes as $q(x_0)\prod q(x_t|x_{t-1})$, we can also rewrite it in reverse as $q(x_T)\prod q(x_{t-1}|x_t)$, and, therefore, it makes sense to approximate it with a Markovian reverse process $p_\theta(x_T)\prod p_\theta(x_{t-1}|x_t)$ (by using the same factorization and sufficiently flexible distributions, we can in theory reduce the approximation gap to zero). However, when the forward process is star-shaped, i.e. it factorizes as $q(x_0)\prod q(x_t|x_0)$, we can only write it in reverse as $q(x_T)\prod q(x_{t-1}|x_t, x_{t+1}, \dots, x_T)$. We could still try to approximate it with a Markovian reverse process $p_\theta$. It was the first thing we tried, and it fails miserably: see Figure 3 for details. In Appendix 2 we show that such approximation introduces a huge irreducible approximation gap which grows with the number of diffusion steps. Because of that we need to use the reverse process with the full dependency structure $p_\theta(x_T)\prod p_\theta(x_{t-1}|x_t, x_{t+1}, \dots, x_T)$. Fortunately, when the noising distributions come from the exponential family, we can rewrite it as a Markov process over sufficient tail statistics $p_\theta(x_T)\prod p_\theta(x_{t-1}|G_t)$, where $G_t=G(x_t, \dots, x_{t+1})$, so the procedure is still computationally efficient.

---

> > ### Comment · Reviewer_EaR3 · 2023-08-16
> >
> > Thanks for your answer which partly solved my concern. I will slightly raise my rating to borderline accept.

---

### Official Review · Reviewer_mgR2 · 2023-07-06

**Soundness:** 3 good
**Presentation:** 2 fair
**Contribution:** 3 good
**Rating:** 7
**Confidence:** 4

**Summary:**

This paper proposes a non-Markovian diffusion model named the star-shaped diffusion model that generates a sequence of noised image from the original image in the forward process.

This paper studies the theoretical foundation of such new type of model showing that if the forward process is based on a subset of the exponential family of distribution, a tail statistics is sufficient statistics for the reversed process. Based on such a foundation, the reversed process can be conducted.



**Strengths:**

The overall structure of this new diffusion model is new, novel and interesting.

The application of PKD theory is sound.

**Weaknesses:**

Although the model is pretty novel and new, there is a lack of motivation from practical problems -- the experiment are weak and kind of toy. A showcase of the unique application of this model is appreciated.

Lack of comparison of some other non-markovian diffusion on constrained domains such as [1,2]

[1] Learning Diffusion Bridges on Constrained Domains
[2] First Hitting Diffusion Models for Generating Manifold, Graph and Categorical Data

**Questions:**

Is there any ODE-version of this SDE-based model?

**Limitations:**

See above

---

> ### Author Rebuttal · Authors · 2023-08-09
>
> Thank you for the comments! We would like to address your questions below.
>
> **Weakness 1:**
> At this stage our goal was to demonstrate that the model can be successfully applied with a variety of noising distributions rather than perfecting the model for each individual task or finding the best task for the model. In general, we would expect non-Gaussian SS-DDPM to outperform Gaussian DDPM when data (or parts of data) lies on manifolds that naturally support distributions from exponential families, e.g. hyperspheres, p.s.d. matrices, simplexes, etc. For example, data like molecular graphs could combine categorical data (atoms), spherical data (orientations) and positive data (distances).
>
> **Weakness 2:**
> Thanks for pointing out “Learning Diffusion Bridges on Constrained Domains”, we missed it! While we don’t have any experiments that overlap with that paper on hand, we would expect Categorical SS-DDPM to perform similar to D3PM. The results could probably be improved by tuning the noising schedule to the particular tasks. Same point holds for the categorical data experiments with FHDM. Regarding FHDM on geodesic data: we achieve a similar performance on the fire dataset (our result is $-1.26 \pm 0.14$, FHDM reports $-1.24 \pm 0.08$). We will conduct the experiments on other geodesic datasets and add a comparison.
>
> **Question 1:**
> Strictly speaking, for now there is no known direct connection between SS-DDPM and SDE-based models beyond the Gaussian case. However due to the stochasticity in the reverse process, SS-DDPM is indeed similar to SDE-based methods. Due to its close connection to DDPM, we suspect that it should be possible to adapt the DDIM model to SS-DDPM. DDIM is a deterministic version of DDPM that is close to ODE versions of SDE-based models.

---

### Official Review · Reviewer_UBua · 2023-07-09

**Soundness:** 4 excellent
**Presentation:** 3 good
**Contribution:** 3 good
**Rating:** 8
**Confidence:** 4

**Summary:**

The paper presents the Star-Shaped DDPM (SS-DDPM), a general recipe for designing a diffusion model with a noising process lying in a general subset of the exponential family. With a Gaussian noising process, SS-DDPM recovers the DDPM. Diverse experiments on synthetic and practical image and text datasets demonstrate the effectiveness of the proposed SS-DDPM.

**Strengths:**

(1) A general recipe for designing a diffusion model, termed Star-Shaped DDPM (SS-DDPM), is proposed.

(2) The recipe is derived and analyzed in detail, with convincing statistical justifications.

(3) Diverse experiments are conducted to demonstrate the effectiveness of the SS-DDPM.

**Weaknesses:**

(1) Many technical details are given in the supplementary material.

(2) The advantages and disadvantages of the SS-DDPM over the DDPM are not explicitly discussed.

**Questions:**

What are the limitations of the proposed SS-DDPM, when compared with other diffusion models?

---

> ### Author Rebuttal · Authors · 2023-08-09
>
> Thank you for the comments! We would like to address your questions below.
>
> **Weakness (1):**
> We have tried to summarize the most crucial technical details in Section 3. Due to space limitations, we had to put more details in the appendix. Unfortunately, the NeurIPS template makes it more difficult to directly reference the appendix (supplementary material has to be uploaded as a separate file), so we hope that the upcoming arXiv version would be easier to read. Please let us know if particular sections stand out as being confusing, and we would do our best to clarify them in the next revision.
>
> **Weakness (2) and Question (1):**
> SS-DDPM is a direct generalization of DDPM, so many properties of the models are shared. The main advantage of SS-DDPM is being able to use other noising distributions which may be more appropriate to the particular task. The main disadvantage of non-Gaussian SS-DDPM is that it loses many interesting extensions of DDPM like DDIM, DDRM and the connection to SDEs. We feel like it should be possible to adapt these extensions to support SS-DDPM, so it is an interesting opportunity for future work. Also, we found it to be more difficult to come up with sensible noising schedules for SS-DDPM. SS-DDPM is rather sensitive to the noising schedule, so it would be nice to have a better way to choose (or learn) noising schedules. In this work we try to reuse the schedules used by existing models by matching the mutual information between the target clean data and the noisy input of the denoiser. Finally, in non-Gaussian SS-DDPM sampling the tail statistic $G_t$ requires sampling the whole tail $x_t, x_{t+1}, …, x_T$. This is not a big deal in most applications, as sampling these datapoints does not require evaluation of neural networks, but it is still worth noting. In practice it doesn’t increase the training time as DNN evaluation is still the bottleneck.

---

> > ### Comment · Reviewer_UBua · 2023-08-20
> >
> > Thanks for the detailed responses. I will increase my rating from 7 to 8.

---

### Official Review · Reviewer_EZ37 · 2023-07-17

**Soundness:** 3 good
**Presentation:** 4 excellent
**Contribution:** 3 good
**Rating:** 6
**Confidence:** 4

**Summary:**

This paper proposed a star-shaped denoising diffusion probabilistic model (DDPM), which extends DDPM to non-Gaussian noises. As a result, the backward/generative process requires conditioning on tails. The authors then propose an efficient tail conditioning strategy which works when the forward process follows an exponential family with linear parameterization. Duality between star-shaped and Markov diffusion processes are also established which provides theoretical support for star-shaped DDPM for its ability to go beyond Gaussian noises. The effectiveness of star-shaped DDPM is demonstrated on several experiments.

**Strengths:**

1. This paper is written very clearly and well organized.
2. It also provides a general framework for DDPM that can incoporate no-Gaussian noises, where previous efforts are designed for specific noises and not generalizable to other distributions.
3. Duality between star-shaped and Markovian diffusion is established, which shows SS-DDPM and vanilla DDPM are equivalent in the Gaussian case.

**Weaknesses:**

1. While experiments on synthetic data demonstrate the ability of SS-DDPM to deal with non-Gaussian noises, the effectiveness of SS-DDPM on real data needs further verification. The current experiment on real data seems a bit inadequate.

2. Star-shaped denoising diffusion models have been proposed before (Rissanen et al. (2022)), and there is no comparison to this previous method.

**Questions:**

1. How does DDPM perform on the Geodesic data in the experiments?
2. How much improvement can SS-DDPM have over DDPM on real data? The advantage over DDPM seems a bit tiny based on current evaluation. Also, in which case would non-Gaussian noise be useful for DDPMs?
3. Rissanen et al. (2022) also proposed a similar star-shaped DDPM, where the reverse process is Markovian based on the heat equation. How does your method compare to their's?

**Limitations:**

Yes, they do.

---

> ### Author Rebuttal · Authors · 2023-08-09
>
> Thank you for the comments! We would like to address your questions below.
>
> **W1 and Q2:**
> DDPM is a special case of SS-DDPM, so it is safe to expect SS-DDPM to perform at least as good as DDPM. At this stage our goal was to demonstrate that the model can be successfully applied with a variety of noising distributions rather than perfecting the model for each individual task or finding the best task for the model. Careful tuning of the noising schedule and other improvements to the training procedure (e.g. resampling timestamps using importance sampling, more flexible reverse distributions, more architectural improvements, etc) could be used to improve the performance further, but the added hyperparameters would require a careful ablation and make the experiments a lot more expensive. In general, we would expect non-Gaussian SS-DDPM to outperform Gaussian DDPM when data (or parts of data) lies on manifolds that naturally support distributions from exponential families, e.g. hyperspheres, p.s.d. matrices, simplexes, etc. For example, data like molecular graphs could combine categorical data (atoms), spherical data (orientations) and positive data (distances).
>
> **W2:**
> IHDM heavily relies on blurring rather than adding noise, resulting in very different diffusion dynamics. The small amount of noise (std=0.01) is there for regularization and to prevent the probabilistic model from being degenerate. Due to blurring and low noise there is a lot of information shared between consecutive timestamps, making both the forward and the reverse processes essentially Markovian. While the ELBO looks similar due to a similar factorization of the forward process, the underlying model is very different: our model is much closer to DDPM than to IHDM. We will expand on these differences in the next revision.
>
> **Q1:**
> We have evaluated the NLL on the fire dataset. Our result is $-1.26 \pm 0.14$. For comparison, the results reported in Riemannian Diffusion Models (Huang et al 2022) are $-1.38 \pm 0.05$ for RDM, $-1.24 \pm 0.07$ for RSDE (Riemannian Score-Based) and $0.28 \pm 0.2$ for DDPM (Stereographic SDE). Our model performs similar to RSDE and RDM and outperforms DDPM.
>
> **Q3:**
> IHDM achieves 18.96 FID on CIFAR-10, while our Beta SS-DDPM achieves 3.17 (IHDM uses 200 steps and we use 1000; at 200 steps our FID is 3.45). Also, while it looks like IHDM can use any noise distribution, it is not trivial to generalize it to other domains: one would need to come up with an appropriate notion of blurring.

---

> > ### Comment · Reviewer_EZ37 · 2023-08-16
> >
> > Thanks for the response, and I will keep the score.

---

### Decision · Program_Chairs · 2023-09-21

**Decision:**

Accept (poster)

**Comment:**

Reviewers agree that this paper makes solid conceptual contributions to the literature on diffusion generative models.  In future revisions, please be sure to include the additional conceptual and experimental comparisons that arose in the discussion, and to clarify the technical points about which reviewers had questions.